# Profiling cell envelope-antibiotic interactions reveals vulnerabilities to β-lactams in a multidrug-resistant bacterium

Andrew M. Hogan [1], A. S. M. Zisanur Rahman[1,4], Anna Motnenko [1,4], Aakash Natarajan [1], Dustin T. Maydaniuk [1], Beltina León [2], Zayra Batun [1], Armando Palacios[1], Alejandra Bosch [2] & Silvia T. Cardona [1,3] ✉

The cell envelope of Gram-negative bacteria belonging to the *Burkholderia cepacia* complex (Bcc) presents unique restrictions to antibiotic penetration. As a consequence, Bcc species are notorious for causing recalcitrant multidrug-resistant infections in immunocompromised individuals. Here, we present the results of a genome-wide screen for cell envelope-associated resistance and susceptibility determinants in a *Burkholderia cenocepacia* clinical isolate. For this purpose, we construct a high-density, randomly-barcoded transposon mutant library and expose it to 19 cell envelope-targeting antibiotics. By quantifying relative mutant fitness with BarSeq, followed by validation with CRISPR-interference, we profile over a hundred functional associations and identify mediators of antibiotic susceptibility in the Bcc cell envelope. We reveal connections between β-lactam susceptibility, peptidoglycan synthesis, and blockages in undecaprenyl phosphate metabolism. The synergy of the β-lactam/β-lactamase inhibitor combination ceftazidime/avibactam is primarily mediated by inhibition of the PenB carbapenemase. In comparison with ceftazidime, avibactam more strongly potentiates the activity of aztreonam and meropenem in a panel of Bcc clinical isolates. Finally, we characterize in Bcc the iron and receptor-dependent activity of the siderophore-cephalosporin antibiotic, cefiderocol. Our work has implications for antibiotic target prioritization, and for using additional combinations of β-lactam/β-lactamase inhibitors that can extend the utility of current antibacterial therapies.

Antimicrobial resistance is a major threat to global public health. In 2019, an estimated 4.95 million deaths were associated with drug-resistant infections[1], and the toll is expected to rise in the future[2]. Gram-negative bacteria consistently top the list as priorities for antibiotic development as they are a leading cause of antibiotic-resistant infections[3].

A significant driver of antibiotic resistance in Gram-negative bacteria lies in the double membrane composition of their cell envelope. The outer membrane is an asymmetric bilayer composed of phospholipids on the inner leaflet and lipopolysaccharide (LPS), decorated with O-antigen units, on the outer leaflet. Asymmetry is maintained by the action of the Mla pathway, which transports excess

[1]Department of Microbiology, University of Manitoba, Winnipeg, Manitoba, Canada. [2]CINDEFI, CONICET-CCT La Plata, Facultad de Ciencias Exactas, Universidad Nacional de La Plata, La Plata, Buenos Aires, Argentina. [3]Department of Medical Microbiology and Infectious Diseases, University of Manitoba, Winnipeg, Manitoba, Canada. [4]These authors contributed equally: A.S.M. Zisanur Rahman, Anna Motnenko. ✉e-mail: Silvia.Cardona@umanitoba.ca

phospholipids from the outer membrane back to the inner membrane[4]. Together, the inner and outer membranes have orthogonal permeability requirements: small hydrophilic compounds (generally <600 Da[5]) are able to diffuse through water-filled porins in the outer membrane, while hydrophobic compounds are able to diffuse through the inner membrane[6]. The peptidoglycan sacculus is not involved in envelope permeability per se, but rather performs the essential function of maintaining cell shape and structural integrity[7]. Many components of the bacterial cell envelope are essential and have no human homologue, thus are attractive targets for a variety of antibiotics. Moreover, the use of small-molecule potentiators has gained traction as a route to increase membrane permeability and the activity of other antibiotics[8].

Bacteria of the genus *Burkholderia* are notorious for their high levels of intrinsic antibiotic resistance due in part to the unique characteristics of the cell envelope[9]. Among them, the lineage known as the *Burkholderia cepacia* complex (Bcc) are opportunistic pathogens that primarily infect immunocompromised individuals. Some species, such as *B. cenocepacia*, can cause a form of pneumonia and bacteremia known as cepacia syndrome[10]. Near uniform resistance to several antibiotic classes severely limits treatment options[11,12], and eradication protocols often require weeks to months of antibiotic cocktails[13,14]. Furthermore, although new therapies are available to treat the symptoms of cystic fibrosis (e.g. CFTR modulators), there may be limited benefit in pathogen clearance[15], but this has not yet been assessed for Bcc infection.

β-lactams remain as one of the few antibiotic classes effective against Bcc species; however, rates of resistance are over 50%[12,16]. Even combination with β-lactamase inhibitors, such as tazobactam or avibactam, may not restore sensitivity in all isolates[17]. Recently, the siderophore-cephalosporin conjugate antibiotic, cefiderocol, was developed and shown to have potent activity against a variety of Gram-negative pathogens[18,19]. Activity was dependent on the availability of ferric iron in the growth medium and was generally not affected by β-lactamase expression[18,20]. However, there have been no studies into the mechanism and susceptibility to cefiderocol in the Bcc. Thus, under-standing the basis of various forms of β-lactam resistance and identifying new ways to increase β-lactam susceptibility is important for treating Bcc infections.

Here, we performed a genome-wide, transposon mutant screen in *B. cenocepacia* K56-2, a previous epidemic isolate and model for Bcc research, against a wide array of cell envelope-targeting antibiotics. We reasoned that transposon disruption of certain genes would create specific deficiencies in the cell envelope, thus revealing mechanisms of cell envelope-related antibiotic resistance in K56-2. We aimed to uncover chemical-genetic interactions which could be exploited to guide therapeutic approaches. Using BarSeq to quantify mutant fitness in 22 conditions[21], we generated over a hundred functional associations that were validated by CRISPRi. Focusing on the clinically relevant antibiotics, we uncovered that hindering undecaprenyl phosphate recycling increased β-lactam susceptibility and that the synergy of the ceftazidime/avibactam combination relies primarily on inhibition of the PenB β-lactamase. Thus, we proposed more potent combinations of avibactam with aztreonam and meropenem. Additionally, we provide the first detailed characterisation of the activity of the recently developed siderophore-cephalosporin conjugate antibiotic, cefiderocol, in the Bcc. Overall, we demonstrate how the analysis of genome-wide antibiotic-genetic interactions can identify susceptibility determinants and antibiotic combinations that may be exploited to bolster current antibiotic therapies.

## Results

### Rationale for the selected antibiotics

The Gram-negative cell envelope presents both a major permeability barrier to antibiotics and a potential source of antibiotic targets. In particular, Bcc species are known for their impermeable cell envelope (~10 fold less permeable compared to *E. coli*[22,23]), which contributes to extreme resistance to membrane-disrupting detergents and multiple classes of antibiotics[24,25]. Our goal was to probe the mechanisms governing cell envelope-related resistance in *B. cenocepacia* K56-2. We therefore assembled a diverse panel of clinical and pre-clinical antibiotics targeting many aspects of cell envelope biogenesis (Fig. 1 and

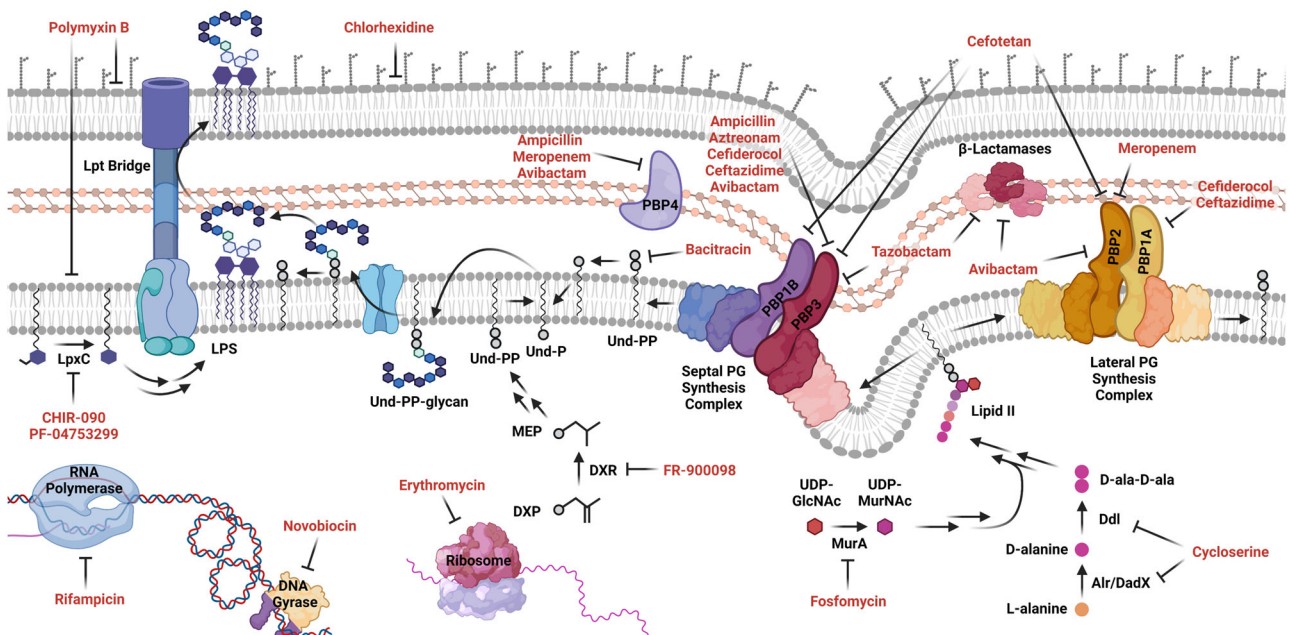

**Fig. 1 | Schematic of the Gram-negative cell envelope and steps inhibited by the antibiotic panel.** Rifampicin, novobiocin, and erythromycin are controls that do not target the envelope. Targets are shown based on previous experimental evidence from *E. coli*, and only if a Bcc homologue exists. High molecular weight PBPs are omitted for clarity. Image created with BioRender.

**Table 1 | Antibiotics used for genome-wide fitness profiling**

| Antibiotic | IC$_{20-30}$ Conc. (µg mL$^{-1}$) | Target (Process) |
|---|---|---|
| Ampicillin (AMP)[b] | 256[b] | PBP3/Divisome (Cell wall synthesis) |
| Aztreonam (AZT) | 40 | PBP3/Divisome (Cell wall synthesis) |
| Cefiderocol (CFD) | 0.04 | PBP3/Divisome (Cell wall synthesis) |
| Ceftazidime (CAZ)[a] | 4 (H); 1.5 (L) | PBP3/Divisome (Cell wall synthesis) |
| Tazobactam (TAZ) | 50 | PBP3/Divisome (Cell wall synthesis) |
| Cefotetan (CTT) | 45 | PBP3/Divisome (Cell wall synthesis) + PBP2/Rod complex (Cell wall synthesis) |
| Meropenem (MEM) | 3.5 | PBP2/Rod complex (Cell wall synthesis) |
| Avibactam (AVI)[a] | 115 (H); 8 (L) | PBP2/Rod complex (Cell wall synthesis) + Diverse β-lactamases |
| Ceftazidime/Avibactam (AVI/CAZ) | 1.5/8 | PBP2/Rod complex (Cell wall synthesis) + Diverse β-lactamases + PBP3/Divisome (Cell wall synthesis) |
| Cycloserine (CYC) | 150 | Alr and Ddl (Cell wall precursor synthesis) |
| Fosfomycin (FOS)[b] | 256[b] | MurA (Cell wall precursor synthesis) |
| Bacitracin (BAC) | 375 | UndPP (Undecaprenol metabolism) |
| FR-900098 (FR-9) | 85 | Dxr (Isoprenoid synthesis) |
| CHIR-090 (CHIR)[b] | 256[b] | LpxC (Lipid A synthesis) |
| PF-04753299 (PF-04) | 0.18 | LpxC (Lipid A synthesis) |
| Chlorhexidine (CHX) | 1.8 | Lipid membranes |
| Polymyxin B (PMB) | 550 | LPS and lipid membranes |
| Erythromycin (ERY) | 100 | 23S rRNA (Protein synthesis) |
| Novobiocin (NOV) | 1.1 | GyrB (DNA supercoiling) |
| Rifampicin (RIF) | 5.5 | RpoB (Transcription) |

[a]Two concentrations of AVI and CAZ were used. A higher concentration (H) that independently reached the IC$_{20-30}$, and a lower concentration (L) the same as that used in the synergistic AVI/CAZ combination. [b]IC$_{20-30}$ could not be reached for AMP, FOS, and CHIR, so highest concentration tested was used (256 µg mL$^{-1}$).

Supplementary Fig. 1A–C; all abbreviations listed in Table 1). We also included three hydrophobic large scaffold antibiotics in this panel that are generally excluded by the Gram-negative cell envelope (rifampicin [RIF], erythromycin [ERY], and novobiocin [NOV]; molecular weight > 600 Da)[5,8]. We expected the large scaffold antibiotics to highlight chemical-genetic interactions in cell envelope permeability and disruptions in major cell envelope biogenesis mechanisms. In summary, we aimed to study cell envelope-associated chemical-genetic interactions and how they may be exploited to inform antibiotic combinations.

### Creating a randomly-barcoded transposon mutant library

To profile genome-wide contributions to antibiotic susceptibility and resistance, we turned to transposon mutagenesis and Illumina sequencing to measure mutant abundance during antibiotic challenges. Previously, our lab has constructed transposon mutant libraries in K56-2, a multidrug-resistant ET12 epidemic lineage clinical isolate[26], to identify the essential genome[27] and characterise targets and mechanisms of action for antimicrobials[28–30]. To leverage advances in sequencing and bioinformatic capabilities, we modified our Tn5 transposon to contain a random 20 bp barcode (Supplementary Fig. 1D), greatly facilitating the tracking of mutant abundance in many conditions by Illumina sequencing[21]. After linking unique barcodes to genomic insertion site (RB-TnSeq), barcodes may be simply amplified by PCR and sequenced (BarSeq) (Supplementary Fig. 1D).

We generated a library of ~340,000 uniquely barcoded transposon mutants in K56-2 (approximately equally divided into 10 subpools). Library statistics can be found in Supplementary Table 1. The library had a median of 12 insertions per protein-coding gene and an average spacing between insertions of 18.7 bp. We found that insertion sites were more likely in low GC-content regions and genes (Supplementary Fig. 2), as we observed previously[27], which may be due to Tn5 transposon insertion and DNA sequencing biases.

To ensure we could use BarSeq to accurately quantify mutant abundance, we performed a pilot experiment with known levels of mutant depletion. Quantification of mutant abundance demonstrated high replicate reproducibility and close agreement with expected levels of depletion (Supplementary Fig. 3A, B). Despite a ~10-fold range in barcode recovery in pools with equal mutant abundance (Supplementary Fig. 3A)—variation that was not due to barcode GC-content (Supplementary Fig. 3C)—we were able to successfully and accurately quantify barcoded mutant depletion at small scale.

### Antibiotic exposure and detection of broad interactions on pathways and processes

The choice of antibiotic exposure conditions has important implications for the sensitivity of high-throughput experiments. Thus, to enable detection of specific mechanism-related interactions, antibiotic doses were selected to inhibit 20–30% of growth relative to growth without antibiotics, as measured by OD$_{600}$[29,31,32] (Table 1 and Supplementary Fig. 4). As controls for the synergistic combination of ceftazidime/avibactam (AVI/CAZ), each component was also used at the lower concentration present in the combination as well as higher concentrations that individually inhibited 20–30% of growth. Ampicillin (AMP), fosfomycin (FOS), and CHIR-090 (CHIR) did not attain 20% growth inhibition so each was used at the highest concentration tested (256 µg mL$^{-1}$).

The entire pool of ~340,000 unique mutants was inoculated in LB medium at OD$_{600}$ 0.025 (~75 CFU per mutant) and allowed to reach early exponential phase (OD$_{600}$ 0.15). The cultures were then exposed to antibiotics (or 1% DMSO solvent control) for 8 h (~10 generations), after which genomic DNA was harvested and used as template for BarSeq. High-output Illumina NextSeq flow cells were used to accommodate ~500 reads per gene for each condition and replicate. Resulting barcodes were counted and matched to insertion site, then

normalized to Time 0 controls and aggregated across replicates to calculate average per-gene fitness scores[21,33].

Comparison between the DMSO control and each condition revealed hundreds of broad and specific factors that contribute to antibiotic susceptibility (Supplementary Fig. 5). As a first look to reveal effects on whole pathways and processes, the genes significantly affecting fitness in each condition were analysed for enrichment in BioCyc pathways and GO terms (Supplementary Figs. 6–8)[34–37]. For negative fitness effects, we observed an enrichment in genes in membrane lipid metabolism pathways (Supplementary Figs. 6–8). Additionally, we found enrichment in many GO terms related to the cell envelope (Supplementary Figs. 7–8). These were, for example, "peptidoglycan metabolic process", "integral component of the membrane", and "periplasmic space". Broadly, these findings are in line with expectations that cell envelope-targeting antibiotics will report on susceptibility determinants in the cell envelope.

Identifying similar hallmarks as other chemical genetics studies would serve as additional validation of our results. For example, although antibiotics act by distinct mechanisms based on their class, many also exert metabolic perturbations and the induction of reactive oxygen species (ROS)[38,39]. We found that disruptions in genes related to purine and pyrimidine metabolism and amino acid metabolism pathways often showed reduced susceptibility to antibiotics, especially the β-lactams (Supplementary Figs. 6–8). Inactivating these genes may slightly reduce metabolic rate and nucleotide pools, both known to affect antibiotic susceptibility[40,41]. Conversely, we found that disruptions in several genes related to ROS mitigation and polyamine and reductant metabolism often increased antibiotic susceptibility (Supplementary Fig. 6). This was also detected by GO term enrichment of "response to oxidative stress" and "glutathione metabolic process" (Supplementary Fig. 7). Glutathione and polyamines, such as putrescine, are both known to be important in *B. cenocepacia* for protection against ROS[42,43]. Together, the corroboration of several previous works and expected findings confirms the validity of our BarSeq approach to uncover detailed chemical-genetic interactions at the foundations of antibiotic action.

## The Mla pathway is important for envelope integrity and resistance to antibiotics

The Mla pathway functions in retrograde transport of excess phospholipids from the outer membrane to the inner membrane, thereby maintaining an asymmetric, and more impermeable, outer membrane enriched in LPS (Fig. 2A)[4,44]. In *B. cenocepacia* and *B. dolosa*, defects in the Mla pathway are known to increase susceptibility to large-scaffold antibiotics and serum[45]. However, homologous defects in *E. coli* K-12 and *P. aeruginosa* PA14 were found to not alter susceptibility to a variety of antibiotics[45]. In K56-2, the Mla pathway is encoded by six genes organized in two operons, and possibly two accessory genes (K562_RS01610 and K562_RS01615) in an adjacent operon (Fig. 2B)[45].

In our BarSeq experiment, genes encoding components of the Mla pathway had negative fitness scores for nearly all tested antibiotic conditions (Fig. 2C and Supplementary Fig. 5). Disruptions in *mlaFED*, generally resulted in greater susceptibility increases, especially for the large scaffold antibiotics (ERY, NOV, and RIF), than did disruption in *vacJ* and *mlaCB* (Fig. 2C). Indeed, the average fitness score of the Mla pathway genes negatively correlated with the molecular weight of the antibiotic (Fig. 2D). This indicates the Mla pathway is important for membrane integrity in K56-2, as large antibiotics are ineffective against Gram-negatives with intact membranes. To support these findings, we used CRISPR-interference (CRISPRi)[46,47] to silence the *mlaFEDvacJ* and *mlaCB* operons. Silencing these genes did not result in a growth defect (Supplementary Fig. 9). Using an *N*-phenyl-1-naphthylamine (NPN) uptake assay[48,49], we found that both of these mutants had substantially increased outer membrane permeability versus the control (Fig. 2E). While the CRISPRi mutant of *mlaFEDvacJ* was expected to be

more permeable than that of *mlaCB*, following the BarSeq findings (Fig. 2C), weaker repression as determined by qRT-PCR (Supplementary Table 2) resulted in nearly equal permeability to NPN. We next reasoned that chemical permeabilization of the membrane may also increase antibiotic susceptibility of K56-2. We found chlorhexidine (CHX), but not polymyxin B (PMB), to greatly increase outer membrane permeability (Supplementary Fig. 10). Consequently, in checkerboard interaction assays, CHX synergised strongly with large scaffold antibiotics and the β-lactams (Fig. 2F). Overall, our findings of broad susceptibility profiles for mutants in the Mla pathway support a unique importance of this pathway in maintaining the permeability barrier of the outer membrane in *Burkholderia*, similar to a previous report[45].

## Antagonising undecaprenyl phosphate recycling causes β-lactam susceptibility, hinders growth, and affects cell morphology

Polysaccharides are important structural and functional components of bacterial cell envelopes, including exopolysaccharides, the O-antigen, peptidoglycan, and protein *O*-glycosylation. From the BarSeq data, we observed that mutants with disruptions in LPS/O-antigen synthesis (e.g. *hldD*, *wbiGH*, *ogcA*, and K562_RS30805) and protein glycosylation (e.g. *ogcABE* and *pglL*) had altered susceptibility to several β-lactam antibiotics (Fig. 3A). To examine if defects in outer membrane permeability were the source of increased antibiotic susceptibility, we assessed NPN uptake of CRISPRi mutants in genes/operons highlighted by the BarSeq experiment. We found that knockdown of genes related to LPS core and O-antigen synthesis and protein glycosylation did not significantly impair outer membrane integrity, except for *hldD* (Fig. 3B). Thus, increased antibiotic susceptibility is not simply due to increased antibiotic influx alone.

We next reasoned that the enhanced susceptibility of cell envelope glycan pathway mutants to β-lactam antibiotics could be due to limitations in a shared intermediate, the lipid carrier UndP (Fig. 3C). As a carrier, UndP is recycled after a cleavage step removes the linked glycans. UndP intermediates are present at low levels in Gram-negative membranes (<1% of total membrane lipids)[50–52] and both de novo UndP synthesis and UndP recycling are essential for viability in *E. coli*[53,54]. Therefore, limiting levels of UndP intermediates has major consequences for the bacterial cell.

Two lines of evidence support a link between UndP utilization pathways, peptidoglycan synthesis, and β-lactam susceptibility. First, disruptions in O-antigen and enterobacterial common antigen pathways render morphological defects in *E. coli* and *Shigella flexneri* similar to those caused by peptidoglycan defects[55–57]. Second, inhibition of de novo UndP synthesis rendered *B. cenocepacia* and *E. coli* more susceptible to β-lactams[58,59]. We thus reasoned that disruptions preventing or reducing the efficiency of UndP recycling may increase susceptibility to the β-lactams, FR-9, and bacitracin (BAC) due to the connection between UndP and peptidoglycan synthesis. Supporting our hypothesis, the BarSeq experiment showed that disruption of *dbcA*, encoding a homologue of the *E. coli dedA* UndP flippase important for UndP recycling[60], resulted in susceptibility to β-lactams, FR-9, and BAC (Fig. 3A). To validate the findings of the BarSeq experiment, we assessed susceptibility of CRISPRi mutants in genes related to cell envelope glycan metabolism to AZT, CAZ, and MEM, as representatives of the β-lactams used in the BarSeq experiment. Susceptibility moderately increased upon knockdown of cytoplasmic steps in the protein *O*-glycosylation pathway (encoded by *ogcABEI*) and LPS core synthesis (encoded by *hldD* and *wabRwaaLwabQP*) (Fig. 3D).

Knockdown of LPS core synthesis genes may cause an accumulation of UndP-O-antigen intermediates in the periplasm as the O-antigen cannot be ligated to a heavily truncated core, thus possibly reducing UndP recycling. To support our genetic evidence for the interactions among UndP utilisation pathways, we used the LpxC

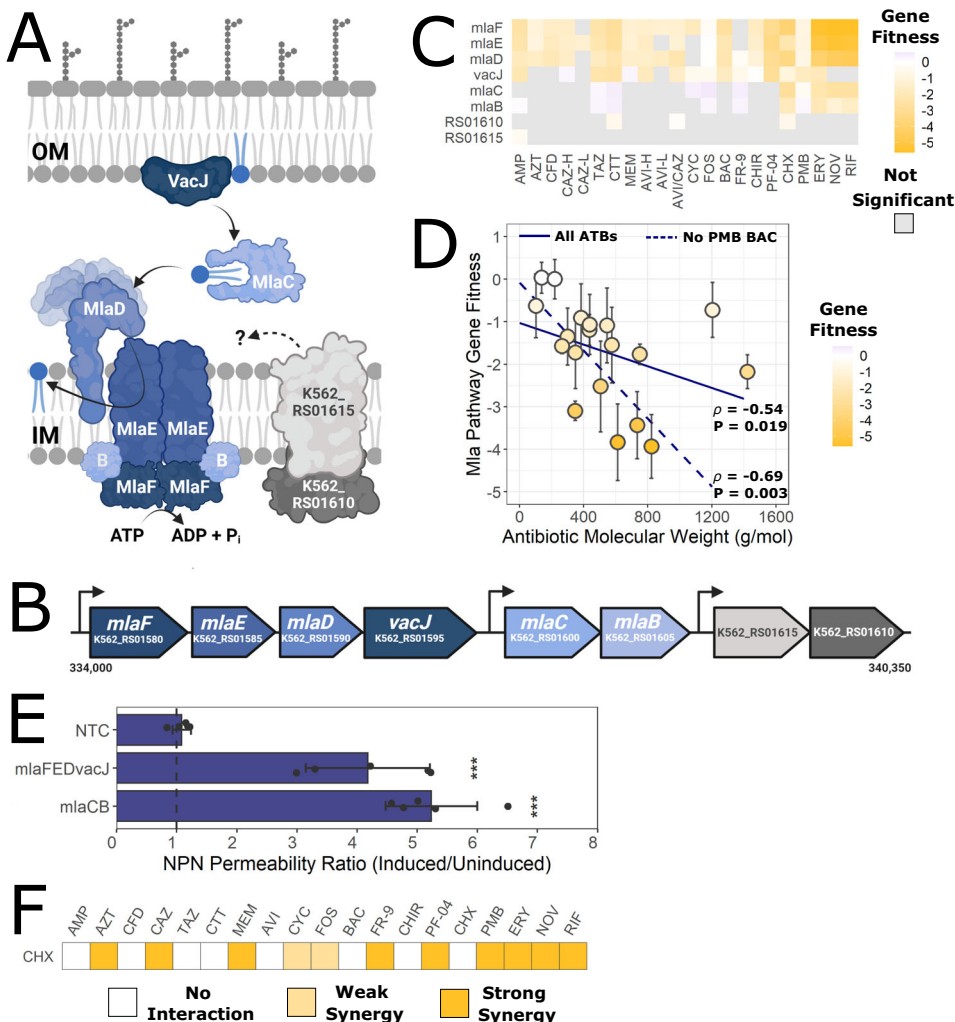

**Fig. 2 | Defects in the Mla pathway increase susceptibility to multiple antibiotics. A** Retrograde phospholipid trafficking from the outer (OM) to inner (IM) membranes by the Mla pathway, inferred by homology[4,45]. The roles of K562_RS01610 and K562_RS01615 are unclear. Image created with BioRender. **B** Genetic arrangement of the Mla genes in K56-2. Nucleotide positions on K56-2 chromosome 1 are noted. **C** Gene fitness scores relative to DMSO control of the genes in the Mla pathway. Grey squares indicate the value was not significant ($P > 0.05$) from a two-sided $t$-test. Further details can be found in the Methods. **D** Correlation of average gene fitness scores in the Mla pathway (*mlaFEDvacJ* and *mlaCB*) from the BarSeq experiment with antibiotic molecular weight. The points are coloured by average Mla pathway gene fitness score from three biological replicates; error bars represent SD. The lines show a linear regression with all

antibiotics (solid) vs. without PMB and BAC (dashed). Shown by each line is the Spearman's rank correlation coefficient ($\rho$) and $P$-value. **E** Ratios of NPN fluorescence (a measure of outer membrane permeability) of the CRISPRi mutants in inducing (0.5% rhamnose) vs uninducing (0% rhamnose) conditions. Error bars represent means ± SD of six biological replicates. Significance was determined by 1-way ANOVA with Dunnett's post hoc test to the non-targeting control sgRNA (NTC). ***$P < 0.001$. Exact $P$-values are $2.7 \times 10^{-6}$ (*mlaFEDvacJ*) and $5.1 \times 10^{-5}$ (*mlaCB*). The dashed line indicates an NPN fluorescence ratio of 1. **F** Summary of antibiotic checkerboard interaction assay with CHX. Interactions were assessed and interpreted with SynergyFinder as per the Methods. Source data are provided as a Source Data file.

---

inhibitor PF-04, which prevents lipid A formation, in checkerboard assays. We reasoned that exposing cells to PF-04 may cause accumulation of UndP-O-antigen intermediates in the periplasm, similar to knockdown of core biosynthetic genes *hldD* and *wabRwaaLwabQP*. Indeed, PF-04 strongly synergised with MEM, CTT, CAZ, AZT, and the isoprenoid synthesis inhibitor FR-9 (Fig. 3E). Interaction with FR-9 suggests PF-04 may cause sequestration of UndP-linked intermediates, while the resulting stress on peptidoglycan synthesis is supported by synergism of PF-04 with the β-lactams. We argue that the observed synergy was not likely due to increased outer membrane permeability as PF-04 did not synergise with the large scaffold antibiotics ERY, NOV, or RIF.

When we silenced LPS core and *O*-glycosylation genes, we noticed that the antibiotic susceptibility profiles were very similar, regardless of the β-lactam used (Fig. 3D). This may indicate an independence from

which peptidoglycan synthesis complex is targeted and instead points to an interaction with UndP levels itself. If UndP sequestration causes β-lactam susceptibility by reducing flux through peptidoglycan synthesis, we expected that reducing the total amount of UndP would produce a similar effect. We then targeted *ispDF* (encoding early genes in isoprenoid/UndP synthesis) and *uppS* (also called *ispU*, encoding UndPP synthase) with CRISPRi. As *uppS* is an essential gene and repression strongly suppresses growth (Supplementary Fig. 9), we carefully titrated the concentration of rhamnose to suppress growth by 20–30% for further assays. Indeed, knockdown of *ispDF* and *uppS* increased susceptibility to AZT, CAZ, and MEM (Fig. 4A). However, knockdown of *ispDF* moderately increased membrane permeability (Fig. 3B), suggesting that the β-lactam susceptibility of that mutant may be partly due to increased antibiotic influx. In addition to genetically depleting levels of UndP, we also investigated antibiotic

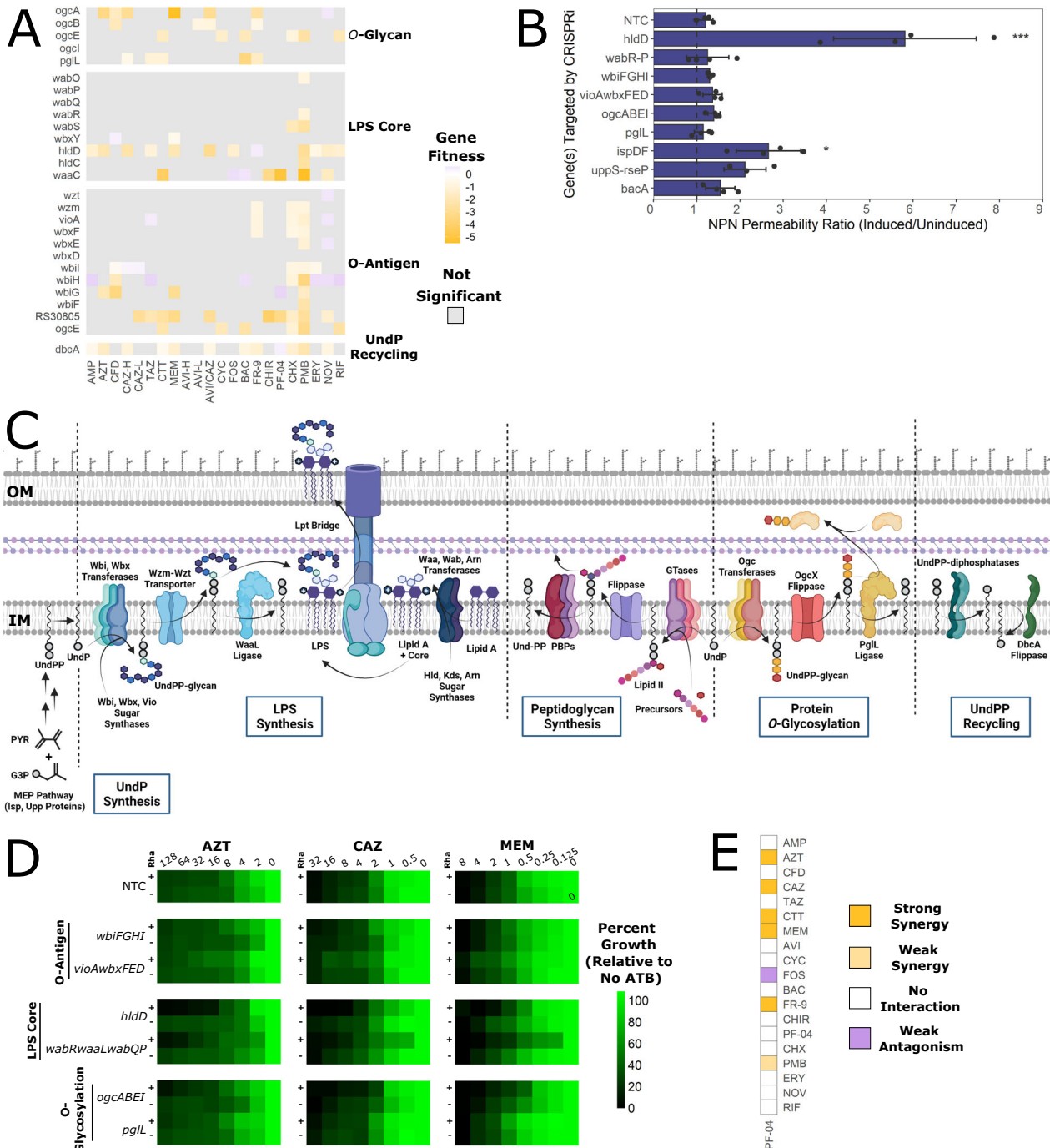

**Fig. 3 | Defects in UndP utilisation pathways sensitize cells to β-lactams. A** Gene fitness scores relative to the DMSO control of genes involved in cell envelope glycan synthesis. Grey squares indicate the value was not significant ($P > 0.05$) from a two-sided *t*-test. Further details can be found in the Methods. **B** Ratios of NPN fluorescence of the CRISPRi mutants in inducing vs uninducing conditions. Error bars represent means ± SD of four biological replicates. Significance was determined by 1-way ANOVA with Dunnett's post hoc test to the non-targeting control sgRNA (NTC). *$P < 0.05$; ***$P < 0.001$. Exact $P$-values are $1.8 \times 10^{-12}$ (*hldD*) and 0.019 (*ispDF*). The dashed lines indicate a NPN fluorescence ratio of 1. **C** Summary of the major UndP(P) metabolic pathways in *B. cenocepacia* (from experimental evidence and inferred by homology), annotated with proteins names if they are known[126–130].

UndPP is synthesized in the cytoplasm by the methylerythritol phosphate (MEP) pathway. UndP is a lipid carrier for construction of the O-antigen, peptidoglycan building blocks (in the form of lipid I and II), and the protein O-glycan. After use as a carrier, UndPP is liberated and recycled into UndP on the cytoplasmic leaflet. IM inner membrane, OM outer membrane, GTase glycosyltransferase. Image created with BioRender. **D** Antibiotic dose responses (µg mL⁻¹) of growth of CRISPRi mutants with or without induction with 0.5% rhamnose. Values are normalized to the OD₆₀₀ of growth without antibiotic and are means of three biological replicates. NTC non-targeting control sgRNA. **E** Summary of antibiotic checkerboard interaction assay with PF-04. Interactions were assessed and interpreted with SynergyFinder as per the Methods. Source data are provided as a Source Data file.

interactions with FR-9 and BAC. FR-9 is an inhibitor of Dxr, which catalyzes an early step in isoprenoid biosynthesis[61], while BAC binds UndPP and prevents its recycling[62] (Fig. 1). FR-9 and BAC synergised with many β-lactams, in addition to with each other (Fig. 4B),

demonstrating that double targeting at different points within UndP metabolic pathways greatly increases the inhibitory effect. Overall, both chemical and genetic evidence supports our assertion that depletion of free UndP pools causes β-lactam susceptibility.

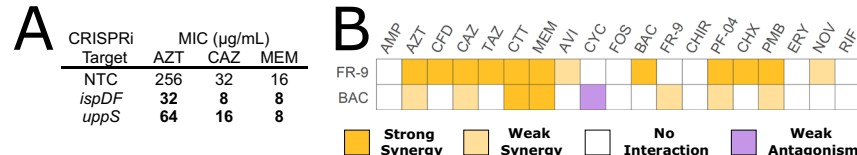

**Fig. 4 | Inhibiting isoprenoid and UndP synthesis sensitizes cells to β-lactams.** **A** MIC values of K56-2::dCas9 harbouring plasmids expressing a non-targeting sgRNA control (NTC) or an sgRNA targeting the indicated genes. MIC values are medians of three biological replicates, with bold indicating at least 2-fold change from the NTC with the addition of rhamnose. **B** Summary of antibiotic checkerboard interaction assay with FR-9 and BAC. Interactions were assessed and interpreted with SynergyFinder as per the Methods. Source data are provided as a Source Data file.

Lastly, to further enhance the block in UndP metabolism, we aimed to simultaneously disrupt UndP synthesis while provoking an accumulation of UndP-linked glycans. These perturbations are expected to severely reduce free UndP pools, and as UndP is an essential molecule for peptidoglycan synthesis, the resulting effects will be synthetic lethality/sickness. To that end, we first separately deleted *hldD*, an epimerase required for LPS core saccharide synthesis, and *waaL*, the O-antigen ligase, in the K56-2::dCas9 background. Mutants of each gene are expected to accumulate UndP-linked O-antigen intermediates. The lack of O-antigen decorated LPS was verified in Δ*hldD* and Δ*waaL* by silver staining LPS extracts, but only the Δ*hldD* mutant could be complemented in trans (Supplementary Fig. 11). We then targeted *ispDF* and *uppS* for CRISPRi knockdown in the Δ*hldD* and Δ*waaL* dCas9 mutants and compared the effects to the K56-2::dCas9 background. Upon induction of dCas9 with rhamnose, knockdown of *ispDF* and *uppS* resulted in a double mutant effect with a further decrease in growth (Fig. 5A). Finally, we overexpressed *wbiI* (encoding an epimerase/ dehydratase required for O-antigen synthesis) and the *wzm-wzt* O-antigen transporter in the Δ*waaL* mutant. We reasoned that overexpression of these genes in the Δ*waaL* mutant may reduce free UndP pools by causing accumulation of periplasmic UndP-O-antigen intermediates. Indeed, overexpression reduced growth more strongly in the Δ*hldD* and Δ*waaL* mutants (Fig. 5B). Additionally, the Δ*waaL*/p*wzm-wzt*+ mutant displayed morphological changes, such as bulging, that are indicative of defects in the peptidoglycan matrix (Fig. 5C). This mutant was also more susceptible to AZT, CAZ, and MEM (Fig. 5D).

Overall, our results support the following model. In wild-type cells, recycling is important to replenish the UndP pools available for the essential process of peptidoglycan synthesis (Fig. 5E). Blockages in UndP utilisation cause sequestration of UndP-glycan intermediates, reducing the efficiency of recycling and the levels of free UndP available for peptidoglycan synthesis. The lack of UndP impaires peptidoglycan synthesis, thus increasing susceptibility to β-lactams.

## BarSeq reveals the basis for β-lactam/avibactam synergy and rationalizes new effective combinations

AVI/CAZ is a current front-line treatment for many infections caused by multidrug-resistant Gram-negative bacteria. The synergy of the combination is due to avibactam inhibiting a broad spectrum of β-lactamases that degrade ceftazidime[63,64]. Studies on *Burkholderia* β-lactamases have focused on the Ambler class A PenB carbapenemase and the Ambler class C AmpC β-lactamase, of which only PenB is inhibited by AVI[65,66]. However, K56-2 encodes a further 19 putative β-lactamase-fold proteins, and it is unknown how/if each contributes to β-lactam resistance, and which may be inhibited by AVI.

In the BarSeq experiment, we saw that transposon disruption of only two β-lactamase genes, *bla*_PenB and K562_RS32470 (encoding a putative metallo-β-lactamase (MBL) fold protein; Pfam 00753) increased β-lactam susceptibility (Fig. 6A). Although K562_RS32470 is annotated as an MBL, the interaction with AZT make us question this assignment as AZT is a poor substrate for most MBLs[67]; however, we did not conduct further experiments to investigate this. Additionally,

disruption of *penR*, the positive regulator of *bla*_penB also increased β-lactam susceptibility (Fig. 6A).

We reasoned that if the targets of AVI are the β-lactamases that degrade CAZ, transposon mutants of said β-lactamases would have a fitness defect in the presence of CAZ because CAZ cannot be degraded. The same mutants would also have a fitness defect in AVI/CAZ as in either case the β-lactamase target is chemically inhibited or genetically disrupted (Fig. 6B). Using the data from our BarSeq experiment, we compared pairs of conditions involved in the AVI/CAZ combination to identify genes important for fitness in one or both constituent conditions. For all comparisons, there were more genes unique to each condition than shared between any two, highlighting strong concentration-dependent physiological effects, even for the same antibiotic (Supplementary Fig. 12). Pair-wise comparison revealed that *bla*_PenB was important for fitness in both CAZ only and AVI/CAZ, while K562_RS32470 was important for fitness in CAZ only (Fig. 6A and Supplementary Fig. 12). None of the other 20 putative β-lactamase-fold proteins were important for fitness in any condition tested here (Supplementary Data 1). Thus, our findings suggest that PenB and K562_RS32470 are the only possible candidate β-lactamase targets of AVI in K56-2.

For validation, we assessed β-lactam susceptibility of CRISPRi knockdown mutants in *bla*_AmpC, *bla*_PenB, and K562_RS32470. In the absence of antibiotics, neither were important for growth (Supplementary Fig. 9). Knockdown of neither *bla*_AmpC nor K562_RS32470 altered the MIC of any of the tested β-lactams (Fig. 6C and Supplementary Table 3). On the other hand, knockdown of *bla*_PenB resulted in marked susceptibility to AMP, TAZ, CAZ, AZT, and MEM (up to 32-fold reduction in MIC) (Fig. 6C and Supplementary Table 3). We reasoned that if PenB is the major β-lactamase in K56-2, then knockdown of *bla*_PenB would result in the same MIC as adding AVI. In other words, the cells would be "blind" to the addition of AVI as the primary target is already knocked down. Indeed, in the presence of AVI, there was no change in MIC upon *bla*_PenB knockdown for CAZ and MEM (Fig. 6C); however, knockdown of *penB* still reduced the MIC of AZT by 2-fold, suggesting that in the absence of PenB, K562_RS32470 may have a minor contribution to AZT resistance. This pattern is also observed when we measured β-lactamase activity using the chromogenic β-lactam derivative nitrocefin (Fig. 6D). Knockdown of *penB* reduced β-lactamase activity by ~90% and was further inhibited to ~98% when AVI was added. There was no effect when K562_RS32470 was knocked down. Unexpectedly, knockdown of *bla*_AmpC increased β-lactamase activity by 13-fold. Although the MIC of the *bla*_AmpC knockdown mutant was not altered, when we re-examined the data in the context of a dose-response, growth at subinhibitory β-lactam concentrations was substantially greater (Supplementary Fig. 13A). As PenB was responsible for most of the β-lactamase activity in our previous assays, we explored if knockdown of *bla*_AmpC may induce *bla*_PenB expression. We thus quantified mRNA expression by qRT-PCR in the *bla*_AmpC CRISPRi mutant and found a 9.1-fold increase in *bla*_PenB expression, but no change in the expression of K562_RS32470 (Supplementary Fig. 13B). Additionally, the heightened β-lactamase activity of the *bla*_AmpC CRISPRi mutant could

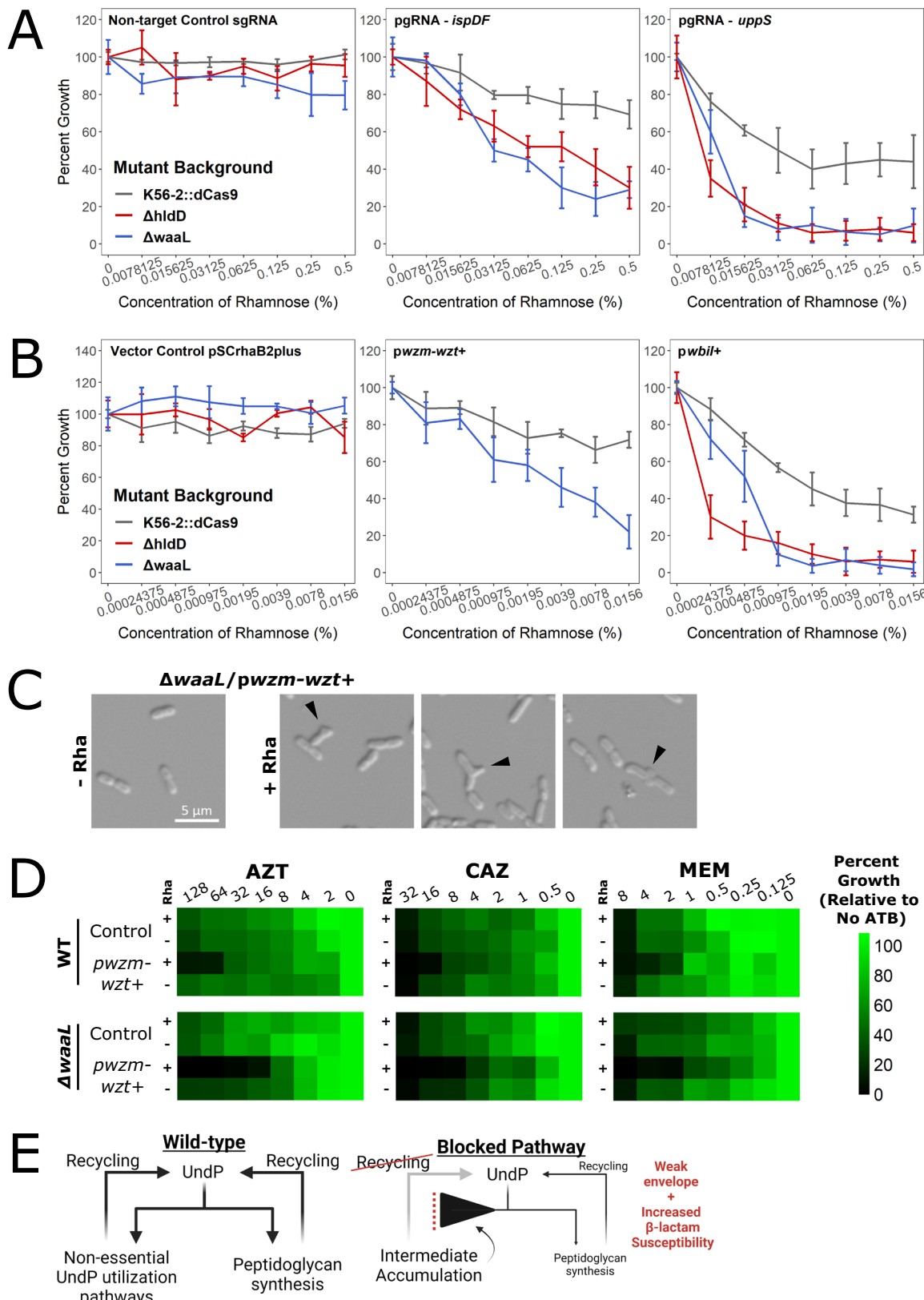

be inhibited by AVI, albeit at very high concentrations (Supplementary Fig. 13C). Taken together, our genetic and biochemical results demonstrate that in K56-2, PenB is the predominant β-lactamase responsible for degrading clinically relevant β-lactams. Our findings also demonstrate that BarSeq can be used to elucidate the mechanisms and targets of antibiotic potentiation.

The marked β-lactam susceptibility of knockdown mutants in $bla_{PenB}$, together with the ability of avibactam to inhibit PenA-family β-lactamases in *Burkholderia* species[68], suggested a broader applicability of AVI/AZT and AVI/MEM combinations. Thus, we assembled a panel of 41 clinical Bcc isolates (including *B. gladioli*) spanning the last two decades and representing the most commonly recovered species

**Fig. 5 | Double mutant effects in envelope glycan metabolism link UndP sequestration to defects in the peptidoglycan matrix.** Cultures of deletion mutants with **A** CRISPRi plasmids or **B** overexpression plasmids were grown with the indicated concentrations of rhamnose for 18 h, the time at which the control strains (non-targeting sgRNA, NTC; vector control, pSCrhaB2plus) reached the maximum $OD_{600}$. Panels are organized by contained plasmid. Values shown are percent growth ($OD_{600}$) relative to growth without rhamnose and are means of three biological replicates ± SD. **C** Mid exponential phase cells of the $\Delta waaL$/p*wzm-wzt*+ mutant were induced with 0.05% rhamnose for 3 h then immobilized on 1% agarose pads and imaged by DIC at 100x magnification. The black arrowheads mark morphological abnormalities such as bulging. Shown are representative images from three biological replicates. **D** Antibiotic dose responses (µg mL⁻¹) of growth of the indicated mutants with or without induction with 0.005% rhamnose. Values are normalized to the $OD_{600}$ of growth without antibiotic and are means of three biological replicates. Control strain harbours pSCrhaB2plus. **E** Proposed framework for how disruptions in UndP metabolism reduce UndP recycling and flux into peptidoglycan synthesis, weakening the peptidoglycan matrix and increasing β-lactam susceptibility. Image created with BioRender. Source data are provided as a Source Data file.

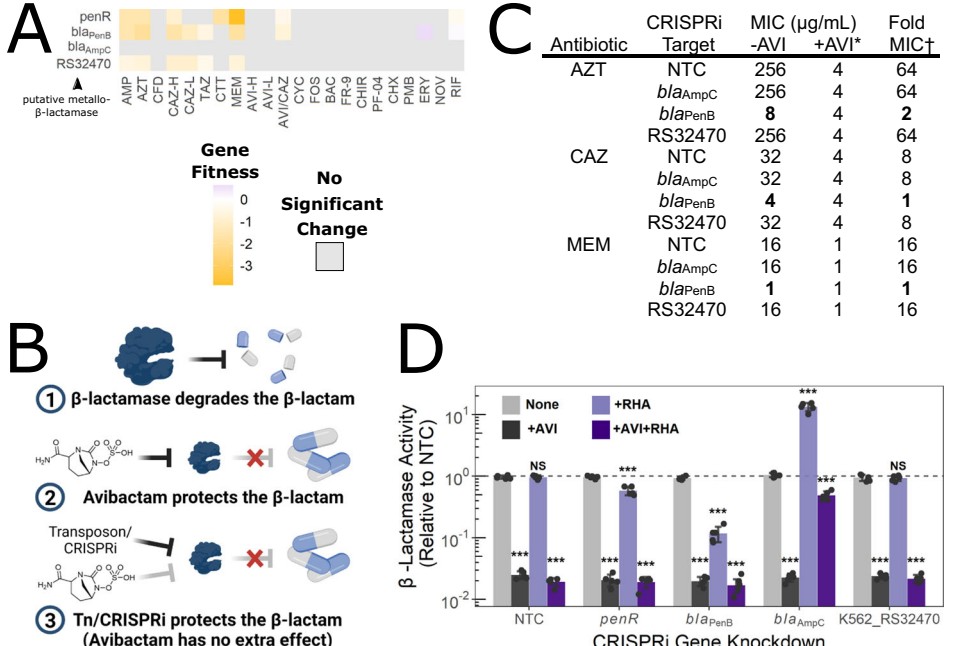

**Fig. 6 | PenB is the predominant β-lactamase in K56-2 and is inhibited by AVI.** **A** Gene fitness scores relative to the DMSO control of β-lactamase and regulatory genes. Grey squares indicate the value was not significant ($P > 0.05$) from a two-sided $t$-test. Further details can be found in the Methods. **B** Rationale for identifying targets of AVI. If a target is disrupted with a transposon or repressed with CRISPRi there will be no change in β-lactam MIC when AVI is added. Image created with BioRender. **C** MIC values of K56-2::dCas9 harbouring plasmids expressing a non-targeting sgRNA control (NTC) or an sgRNA targeting the indicated genes. MIC values are medians of three biological replicates, with bold indicating change versus the NTC. † Fold MIC is the ratio of the MIC -AVI to the MIC + AVI. * AVI kept constant at 8 µg mL⁻¹. **D** Nitrocefin hydrolysis assay of lysate from CRISPRi mutants grown in the indicated conditions. Data are presented as mean values of five biological replicates ± SD, with the dashed line indicating no difference vs. the NTC. Significance was determined by an unpaired two-tailed $t$-test to the NTC grown without rhamnose or AVI using Bonferroni's correction. ***$P < 0.001$. Source data are provided as a Source Data file.

across Canada. Susceptibility to AZT, CAZ, and MEM was assessed with and without 8 µg mL⁻¹ AVI. Among Bcc species, MIC values of the β-lactams alone varied widely: 2 – >256 µg mL⁻¹ for AZT; 0.5 – 32 µg mL⁻¹ for MEM; 2 – >128 µg mL⁻¹ for CAZ (Supplementary Data 2). Overall, potentiation by AVI was strongest for AZT and MEM (up to 64-fold MIC reduction) (Fig. 7A). These trends are in line with the changes in susceptibility upon $bla_{PenB}$ knockdown in K56-2 (Fig. 6C). Consequently, and in the context of clinical breakpoints, 24/41 of the Bcc isolates were resistant to AZT without AVI, which was reduced to 2/41 with AVI (Fig. 7B). For MEM and CAZ, 9/41 and 4/41 of the Bcc isolates were resistant without AVI, respectively, and all Bcc isolates were sensitive with AVI (Fig. 7B).

The activity of AZT, CAZ, and MEM was not uniformly potentiated by AVI in all Bcc isolates. Even for K56-2, in which we have demonstrated the importance of PenB, AVI does not potentiate the activity of all β-lactams or even all cephalosporins (Supplementary Table 4). Moreover, even if AVI potentiated the activity of one β-lactam, it did not guarantee potentiation for the others (Supplementary Data 2). Thus, although alternative β-lactamases that are not inhibited by AVI may contribute to β-lactam resistance, they likely only play minor roles in most isolates. Overall, our findings suggest that, in addition to

AVI/CAZ, combinations of AVI/AZT and AVI/MEM may be valuable therapeutic options for treating Bcc infection.

Our strain panel also included isolates of other CF pathogens: *P. aeruginosa*, *Achromobacter xylosoxidans*, and *Stenotrophomonas maltophilia*. In contrast to the activity against the Bcc isolates, there was minimal potentiation by AVI, except with highly β-lactam resistant *P. aeruginosa* and with AZT against *S. maltophilia* (Supplementary Data 2). As the potentiation of AZT, CAZ, and MEM by AVI was generally weaker in CF pathogens other than the Bcc, this highlights the differences in β-lactamase arsenals and demonstrates the need to perform genome-wide investigations in each species to uncover resistance mechanisms. Consequently, we suggest that among CF pathogens, combinations of AVI/AZT, AVI/CAZ, and AVI/MEM may be more tailored for use against Bcc infections.

## Cefiderocol uptake is via several TonB-dependent receptors and requires physiological levels of iron for activity

Cefiderocol (CFD) is a recently-developed antibiotic with a catechol siderophore conjugated to a cephalosporin that is structurally similar to CAZ. The siderophore chelates ferric iron and enables active transport into cells by TonB-dependent receptors (TBDRs), resulting in

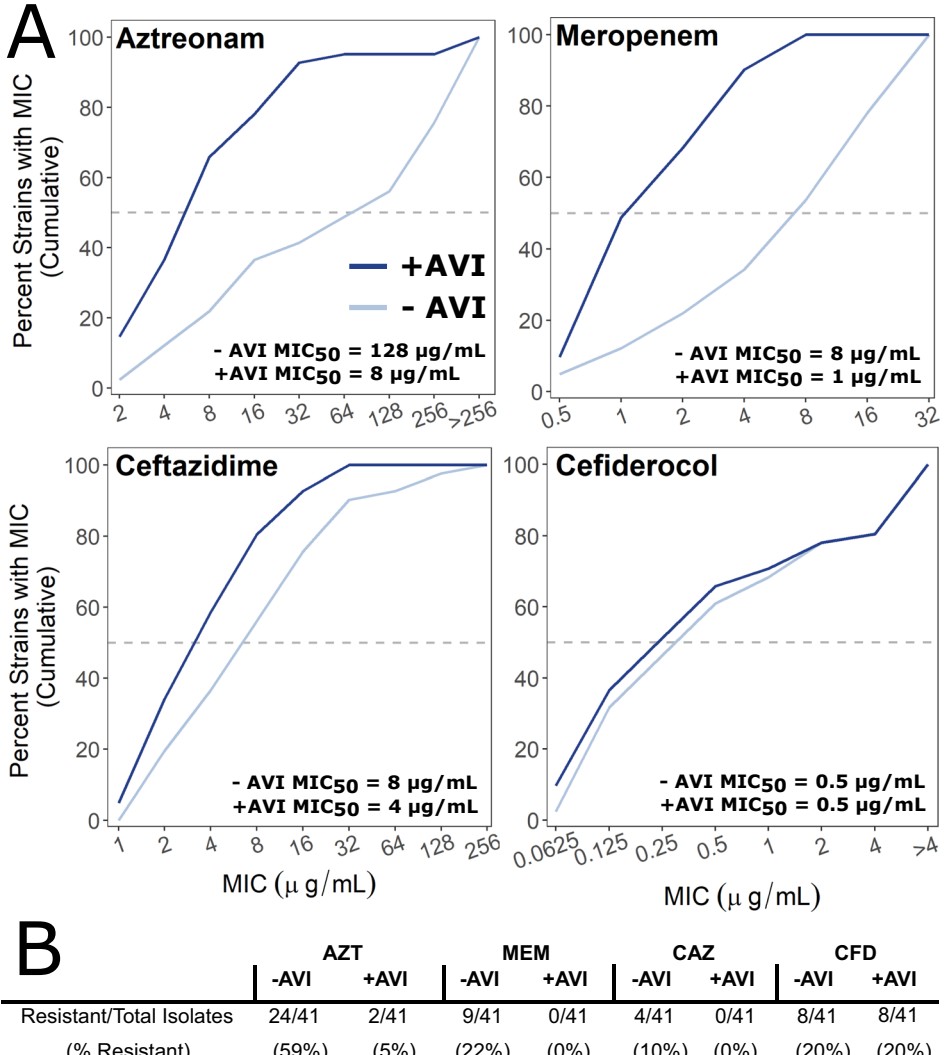

**Fig. 7 | AVI strongly potentiates AZT and MEM in Bcc clinical isolates. A** Percent cumulative growth inhibition of 41 Bcc clinical isolates with or without AVI. The dashed line indicates the $MIC_{50}$, the concentration representing the MIC for 50% of the isolates. Data for panels A and B were summarized from Supplementary Data 2. **B** Resistance of Bcc clinical isolates to AZT, MEM, CAZ, and CFD with or without

8 µg mL$^{-1}$ AVI. CLSI resistance breakpoints for the Bcc: MEM ≥ 16 µg mL$^{-1}$; CAZ ≥ 32 µg mL$^{-1}$. CLSI resistance breakpoints for *P. aeruginosa* were used for AZT (≥32 µg mL$^{-1}$) and CFD (>4 µg mL$^{-1}$) as none exist for the Bcc. Source data are provided as a Source Data file.

substantially increased potency against a variety of Gram-negative pathogens[18,69,70]. Little is known about the activity of CFD in *Burkholderia* species, except for a few cases of cefiderocol use in compassionate care[71] and as part of large strain panels[19,72].

To explore the antibiotic mechanism of CFD, we analysed our BarSeq data for chemical-genetic interactions specific to CFD activity. Broadly, interactions were enriched in pathways related to iron and heme metabolism, which was not observed for other β-lactams (Supplementary Figs. 6–7). Additionally, disruptions of β-lactamase genes $bla_{PenB}$ and K563_RS32470 did not affect fitness in CFD (Figs. 6A and 8A). However, disruptions in K562_RS04910 (encoding a TonB-related protein) and K562_RS23150 (encoding a homologue of the *P. aeruginosa* PAO1 *piuA* TBDR) specifically reduced susceptibility to CFD (Fig. 8A). PiuA is likely a transporter of catechol-type xenosiderophores in *P. aeruginosa*[73]. Of 24 putative TBDRs encoded by K56-2, disruption of *piuA* was the only TBDR associated with significantly enhanced fitness in the rich medium conditions of the BarSeq experiment (Supplementary Data 1). We confirmed the involvement of *piuA* in CFD activity with a CRISPRi knockdown, which showed reduced susceptibility to

CFD (4-fold increase in MIC) but not to the structurally related CAZ (Fig. 8B).

Iron acquisition mechanisms, such as TBDRs, are generally upregulated in low iron conditions[74]. In *P. aeruginosa*, the susceptibility to CFD increases when iron is limited[70]. To test the effect of iron levels on susceptibility of K56-2 to CFD, we used a low iron medium (M9 salts + casamino acids [M9 + CAA]) and high iron medium (CAMHB), which we found by ICP-MS to have 0.61 µM ± 0.07 µM and 6.75 µM ± 0.71 µM total iron, respectively. Contrary to what was observed in *P. aeruginosa*, the MIC of CFD was 8-fold higher in M9 + CAA than in CAMHB for K56-2 (Fig. 8B). This trend was also seen in many Bcc clinical isolates (Supplementary Table 5). Furthermore, lowering iron levels by chelation antagonised CFD in both CAMHB and M9 + CAA (Fig. 8C). While we thought that changes in susceptibility between high and low iron media may be due to reduced *piuA* receptor expression, knocking down *piuA* in M9 + CAA caused a further 4-fold increase in MIC (Fig. 8B). Thus, unexpectedly, low levels of iron reduced the activity of CFD in K56-2 and more broadly across Bcc species.

While infection settings are generally iron-limiting[75], the sputum of individuals with severe CF is enriched in iron (62–125 µM compared

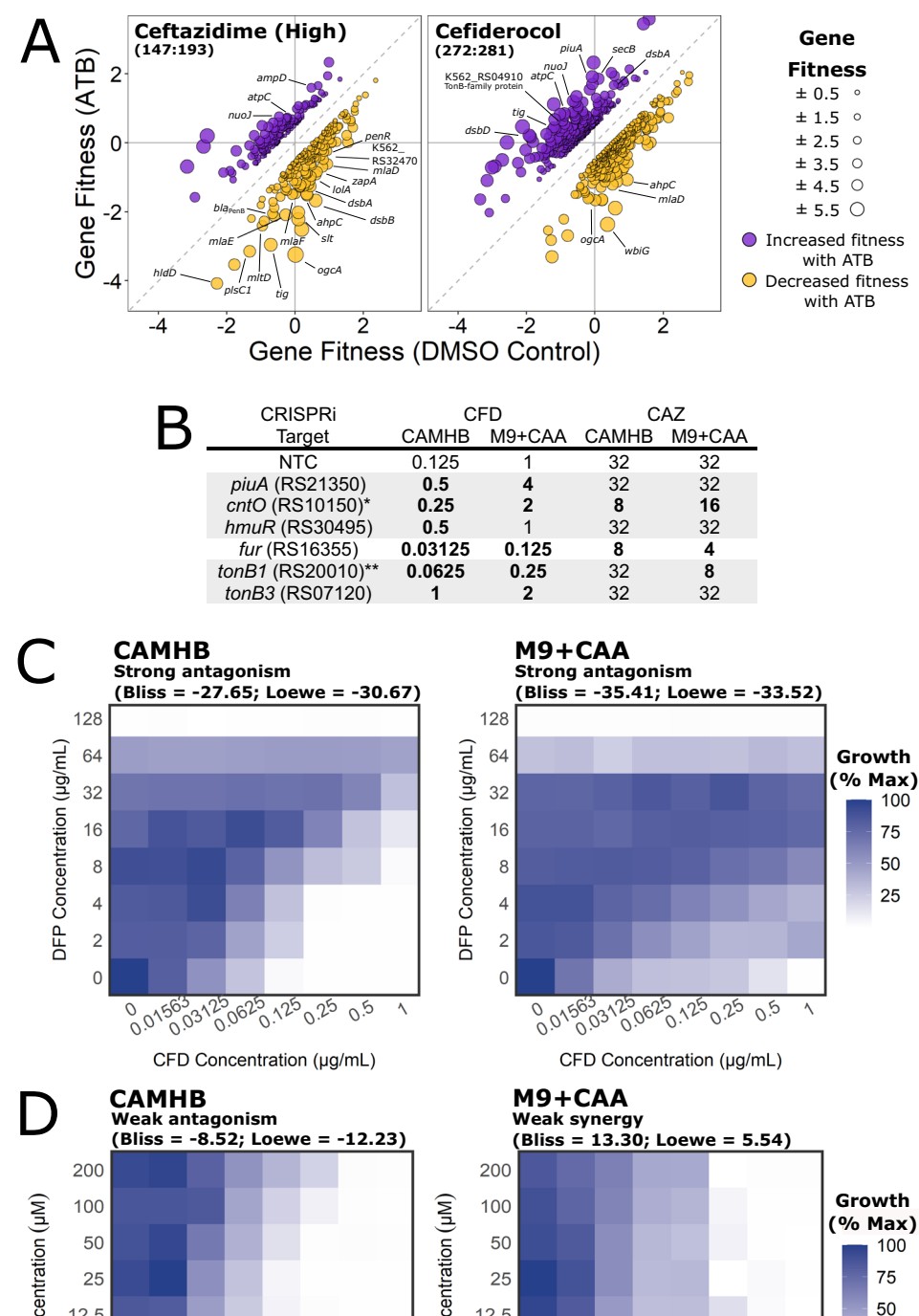

**Fig. 8 | CFD uptake in K56-2 is strongly dependent on growth medium iron concentration. A** Gene fitness profiles of CAZ and CFD. The points are coloured based on positive (increased fitness; purple) or negative (decreased fitness; gold) interactions. All fitness profiles are found in Supplementary Fig. 5. **B** Susceptibility of CRISPRi mutants in genes associated with iron uptake to cefiderocol and ceftazidime in CAMHB and M9 + CAA. MIC values (µg mL⁻¹) presented are medians of at least four biological replicates in the presence of 0.5% rhamnose. Bold indicates at least 2-fold difference from the non-targeting sgRNA control (NTC). Gene names are assigned by homology, if known, or else the locus tag is given (preceded by K562_). *K562_RS10150 was assigned as a strong homologue to *P. aeruginosa* PAO1 *cntO* but failed reciprocal best-hit BLAST. ** *tonB* paralogues are assigned based on similarity to the *P. aeruginosa* PAO1 *tonB1*. **C** Growth of K56-2 in a 2-dimensional antibiotic gradient of CFD and deferiprone in CAMHB and M9 + CAA. **D** The same as for panel B, but with CFD and FeCl₃. Values are normalized to the OD₆₀₀ of the well with the most growth. Interactions were assessed and interpreted with Synergy-Finder. Source data are provided as a Source Data file.

to ~18 μM in a healthy lung)[76]. To mimic the high iron conditions of the CF sputum, we supplemented the CAMHB and M9 + CAA media with FeCl₃. Supplementing M9 + CAA with 25 μM iron resulted in 4-fold lower MIC values (Fig. 8D). At very high iron concentrations (>100 μM) in CAMHB, the MIC was 4-fold higher (Fig. 8D). These effects reflect the different initial iron concentrations in rich CAMHB and defined M9 + CAA, where adding small amounts of iron equilibrate CFD susceptibility between CAMHB and M9 + CAA. These findings are in agreement with the importance of iron for CFD susceptibility.

We then reasoned that the BarSeq experiment, which was performed in rich LB medium, may not have captured the physiological changes that occur in low iron conditions, such as altered TBDR expression. To further study the link between CFD susceptibility and cell physiology under iron limitation, we constructed a panel of CRISPRi mutants in all 24 putative TBDRs encoded in K56-2, in addition to *fur*, encoding the iron-responsive Fur transcriptional repressor, and the four *tonB* paralogues (Supplementary Table 6 and Supplementary Fig. 9). We identified two other receptors, the HmuR heme receptor homologue (K562_RS30495) and a putative CntO pseudopaline receptor homologue (K562_RS10150), that upon knockdown reduced susceptibility to CFD. Additionally, knockdown of the *tonB3* homologue also reduced susceptibility to CFD (Fig. 8B). Knockdown of the *tonB1* paralogue, however, produced opposite effects, increasing susceptibility (Fig. 8B). Knockdown of *fur* increased CFD susceptibility at least 4-fold in both CAMHB and M9 + CAA, suggesting that CFD receptor gene expression is repressed by Fur.

Notably, K56-2 was much more susceptible to CFD at all iron concentrations than to the other β-lactams, even when in combination with AVI (Supplementary Data 2). In addition, the Bcc clinical isolate panel showed the same trends. Of the 41 isolates, the MIC₅₀ was 0.5 μg mL⁻¹ and only eight (20%) were resistant to CFD (Fig. 7B and Supplementary Data 2), demonstrating very potent activity. Susceptibility of other Bcc isolates to CFD was generally not affected by the addition of AVI (Supplementary Data 2). Overall, our findings indicate that while there might be a critical threshold of iron required for CFD activity, CFD MIC values remain well below those of other β-lactams.

### Visualization of chemogenomic interactions in a web application

To ease data accessibility, we have made an interactive and visual web app accessible at https://cardonalab.shinyapps.io/bcc_interaction_viewer/. The effects of individual antibiotics can be filtered with customizable gene fitness thresholds and assessed with output tables that display known annotations separated by interactions that either increase or decrease fitness in the selected condition. Additionally, we can update this web app to hold data from future screens. We expect this user-friendly application will add value to our dataset by allowing additional data mining.

## Discussion

The alarming rate of antibiotic resistance among Gram-negative organisms requires immediate attention. NGS-based assays provide scalable high-throughput approaches that can be applied to virtually any organism. The chemical-genetic interaction profile output can be examined to inform on antibiotic resistance and susceptibility determinants to guide rational antibiotic combinations. Here, using *B. cenocepacia* K56-2 as a model, we measured genome-wide fitness contributions in the presence of 22 antibiotic conditions primarily targeting the cell envelope. Generation of thousands of chemical-genetic interactions allowed confident assignment and validation of over a hundred functional associations for many genes not previously associated with antibiotic susceptibility in *Burkholderia*. By screening with compounds that target the cell envelope, the majority of the identified factors play roles in cell envelope biogenesis and represent approaches to overcome antibiotic resistance.

The centrality and essentiality of UndP for cell envelope biogenesis hints at a weakness that may be exploited for antibiotic development. In *B. cenocepacia*, at least four pathways use UndP intermediates (Fig. 3C, plus the 4-amino-4-deoxy-L-arabinose [Ara4N] lipid A modification), and two are essential for growth (peptidoglycan synthesis and the Ara4N modification). Our BarSeq experiment pointed to disruptions in lipid A/O-antigen synthesis and protein *O*-glycosylation causing susceptibility to β-lactams, FR-9, and/or BAC. Targeted genetic and chemical validations indicated midpoint disruptions in these pathways caused a buildup of UndP intermediates, thus reducing the free UndP available for peptidoglycan synthesis. Mutations causing accumulation of UndP-O-antigen and UndP-enterobacterial common antigen intermediates in *E. coli*[55,56] and *Shigella flexneri*[57] are known to cause substantial morphological defects from insufficient peptidoglycan synthesis. While β-lactam susceptibility has been observed in large-scale screens for disruptions in UndP utilisation pathways in *A. baylyi*[32], *Vibrio cholerae*[77], *P. aeruginosa*[78,79], and *E. coli*[80], we are the first to validate these interactions with targeted genetic and chemical investigations.

Integrity of the cell envelope is monitored and maintained by several stress response pathways. Deficiencies in undecaprenyl phosphate pools[81] and peptidoglycan matrix structure[82] are known to activate envelope stress responses, such as the Rcs phosphorelay and Cpx pathway, in *E. coli*. Among the products of envelope stress response activation is altered antibiotic susceptibility[83]. *B. cenocepacia* K56-2 apparently lacks known homologues of the Rcs and Cpx pathways, and only the σᴱ (RpoE) stress response has been characterized[84] in this species. We observed no substantial changes in β-lactam susceptibility for elements of the σᴱ response or regulon in our dataset, suggesting there is no direct link between UndP and the σᴱ response in K56-2. However, we cannot discount that other uncharacterized stress response pathways may be contributing to our observed phenotypes by sensing and responding to levels of undecaprenyl phosphate or peptidoglycan matrix defects.

An important question that remains is if targeting UndP utilisation pathways is a viable strategy for antibiotic development. Indeed, there are two well-documented cases: targocil and BTZ043[85–87]. Targocil is an inhibitor of the TarG transporter for wall teichoic acid (WTA), a surface-linked glycan constructed on UndP, in *Staphylococcus aureus*. However, as the WTA pathway is non-essential, spontaneous resistance occurs relatively frequently in vitro (~1 in 10⁶ cells)[87]. BTZ043 is an antimycobacterial that targets DprE1 in the essential cell wall arabinan biosynthesis[86]. In *Corynebacterium glutamicum*[88], BTZ043 was found to inhibit growth via sequestering decaprenyl phosphate intermediates required for peptidoglycan synthesis[89]. Overall however, the outlook is not favourable for targeting UndP utilisation pathways in Gram-negatives, as suppressor mutants are commonly identified when attempting to manipulate these non-essential pathways[56,90–92].

Furthermore, while single genetic mutations in UndP utilisation pathways in *E. coli* substantially affect cell morphology[55,56,92], we did not see this in K56-2 (data not shown). It was only upon deleting the *waaL* O-antigen ligase and overexpressing the *wzm-wzt* O-antigen transporter that cell morphology displayed evidence of peptidoglycan cell wall defects (Fig. 5C). *Burkholderia* species may be more robust to defects in UndP metabolism perhaps because have higher initial abundance of UndP. Synthesis of Ara4N requires one UndP carrier each[50], and as most LPS molecules in the *Burkholderia* outer membrane are constitutively functionalized with at least one Ara4N[93], this modification may otherwise place a very high burden on UndP pools. Furthermore, the specific essentiality of the Ara4N modification in *Burkholderia* species[27] supports higher UndP pools to limit consequences on envelope biogenesis.

In addition to developing new antibiotics, there is also a great deal of interest in discovering potentiators (also called adjuvants) that synergise with antibiotics to substantially increase potency. The

presented transposon-based screen is well-suited to identify attractive targets for potentiators, as gene disruption/silencing can mimic the effect of gene product inhibition with a small molecule[94]. In this context, the fitness profile of the mutant in *dbcA* deserves further attention. Disruption of *dbcA* increased susceptibility to β-lactams, BAC, FR-9, CHX, and PMB (Fig. 3E). DbcA, a homologue of the DedA UndP flippase in *E. coli*[95,96], was found to be important for the Ara4N lipid A modification in *B. thailandensis*[93], and a Δ*dbcA* mutant had a colistin MIC of 8 μg mL⁻¹ (compared to > 256 μg mL⁻¹ in WT). The peak sputum concentration of some inhaled colistin therapies is above 300 μg mL⁻¹ sputum[97], well above the inhibitory concentration for a mutant lacking DbcA. It is tempting to suggest that DbcA, and UndP recycling more broadly, may be a linchpin in both β-lactam and cationic antibiotic resistance in *Burkholderia*. Thus, inhibiting UndP recycling with a small molecule may potentiate the activity of multiple clinically available antibiotics.

The activity of β-lactams can be potentiated by β-lactamase inhibitors, such as AVI and TAZ, by protecting them from hydrolysis by β-lactamases. AVI/CAZ shows promise as a treatment for Bcc respiratory infection[98,99]; however, resistance has been identified to this combination[17]. By comparing fitness profiles of CAZ treatment in the presence or absence of AVI, we identified that the PenB serine β-lactamase and the K562_RS32470 putative metallo-β-lactamase are targets of AVI. Targeted validations then demonstrated that PenB is the predominant β-lactamase in K56-2 and primary target of AVI. This was despite qRT-PCR analysis showing no CRISPRi silencing effect in the *bla*PenB mutant, which we attribute to assessing expression without β-lactamase exposure[66]. A similar transposon-based approach was recently used by the Manoil group in *A. baylyi* to show that AVI potentiates CAZ by inhibiting the GES-14 β-lactamase, while MEM is potentiated by AVI inhibition of GES-14 and OXA-23[32]. During our follow-up experiments with PenB, we discovered that knocking down *bla*AmpC strongly induced β-lactamase activity and *bla*PenB expression in K56-2 (Fig. 6D and Supplementary Fig. 13). The *B. multivorans* ATCC17616 AmpC has been shown to degrade β-lactams very slowly, leading Becka et al. to propose it plays a minor role in resistance in *B. multivorans*[66]. We build on this and suggest that the *Burkholderia* AmpC may in fact play a role in peptidoglycan recycling, perhaps by degrading the muropeptide intermediates that activate PenR; however, this requires further study.

We then exploited the dependence on PenB to demonstrate that AVI strongly potentiates AZT and MEM in a panel of Bcc clinical isolates (Fig. 7 and Supplementary Data 2). These combinations are not commonly reported for treatment of Bcc infection and may represent alternative therapeutic approaches. AVI/AZT is reported to be most potent against species of enterobacteria that express a serine β-lactamase (which is inhibited by AVI) and a metallo-β-lactamase (which generally cannot degrade AZT)[100]. However, for treatment of CF respiratory infections, inhaled formulations of aztreonam are available that can achieve concentrations over 725 μg g⁻¹ sputum with new inhaler technology[101], which is nearly 100-fold the MIC50 we determined for the AVI/AZT combination. AZT is also generally-well tolerated by people with sensitivity to other β-lactams[102]. Although inhaled MEM has been used in some cases[103], no other β-lactams are currently licensed as inhaled formulations.

Our genome-wide view into the mechanism of action of CFD may have an impact in moving CFD forward for therapeutic uses as it is not yet indicated to treat CF infections. We identified three receptors for CFD uptake in K56-2: PiuA (K562_RS21350), HmuR (K562_RS30495), and K562_RS10150 (putative CntO homologue). The CRISPRi mutant panel of TBDRs we constructed may also be useful for screening other small molecules that enter the cell via TBDRs, as recently demonstrated for thiostrepton[104] and ubonodin[105]. Promisingly, we also found that CFD activity was not affected by disruptions

in the major β-lactamases (Fig. 6). Additionally, in the largest screen of CFD activity in Bcc isolates to date, 85% were susceptible to CFD alone and there was little benefit of combination with AVI (Fig. 7). Indeed, CFD is generally stable to β-lactamase hydrolysis; however, the NDM-1 and NDM-5 metallo-β-lactamases are linked to resistance in *Enterobacter cloacae* and *Klebsiella pneumoniae*, respectively[106,107]. An additional consideration for CFD activity is also the concentration of iron in the growth conditions. We observed that physiological iron concentrations, even elevated levels associated with chronic CF[76], are adequate for CFD activity. Furthermore, CFD activity was reduced by iron limitation from growth in a low-iron medium and from chelation, effects that we attribute to activity of the Fur ferric uptake regulator. We therefore suggest that critical iron concentration thresholds exist that reflect the interplay between the availability of free iron for CFD to bind and the cellular responses to iron levels. In summary, we demonstrate how chemogenomics can be used to functionally annotate uncharacterised genes and prioritize targets for rational antibiotic combinations to increase effectiveness of current antibiotic therapies.

## Methods

### Strains, medium preparation, and growth conditions

All strains used in this study are listed in Supplementary Table 7. Growth at 37 °C with 230 rpm shaking in LB-Lennox medium (Difco) was used for routine strain propagation for all strains, unless otherwise specified. Cation-adjusted MHB medium (Oxoid) for susceptibility testing was prepared as per manufacturer recommendations. The M9 + CAA medium used for some susceptibility tests consisted of the following: 1x M9 salts (Difco), 25 mg L⁻¹ CaCl₂, 12.5 mg L⁻¹ MgCl₂, 0.3% low-iron casamino acids (Fisher), and 25 mM glycerol. All plasmids used in this study are listed in Supplementary Table 8. The following selective antibiotics were used: trimethoprim (Sigma; 50 μg mL⁻¹ for *E. coli* and 100 μg mL⁻¹ for *B. cenocepacia*), kanamycin (Fisher Scientific; 40 μg mL⁻¹ for *E. coli*), gentamicin (Fisher Scientific; 50 μg mL⁻¹ for *B. cenocepacia*), and tetracycline (Sigma; 20 μg mL⁻¹ for *E. coli* and 100 μg mL⁻¹ for *B. cenocepacia*)

Stock solutions of D-glucose (Fisher Scientific) and L-rhamnose (Sigma) were made to 20% in de-ionized water then filter sterilized. The following antibiotics were stocked in sterile de-ionized water: kanamycin sulphate (Fisher), gentamicin sulphate (Fisher), ampicillin sodium (Sigma), ceftazidime hydrate (Sigma), ceftriaxone sodium (Sigma), cefotaxime sodium (Sigma), cephalexin hydrate (Carbosynth), cefmetazole (Sigma), cefoperazone sodium (Sigma), moxalactam sodium (Sigma), polymyxin B sulphate (Sigma), colistin sulphate (GoldBio), novobiocin sodium (Sigma), and FR-900098 sodium (Toronto Research Chemicals). The following antibiotics were stocked in 100% DMSO (Fisher): aztreonam (Alfa Aesar), cefiderocol (MedKoo Biosciences), trimethoprim (TCI America), tazobactam sodium (Alfa Aesar), cefotetan sodium (Target Mol), meropenem trihydrate (MedKoo Biosciences), avibactam sodium (MedKoo Biosciences), CHIR-090 (MedKoo Biosciences), PF-04753299 (Sigma), chlorhexidine hydrochloride (Angene China), and rifampicin (Sigma). The following antibiotics were stocked in 95% ethanol: chloramphenicol (Sigma), tetracycline hydrochloride (Sigma), and erythromycin (Sigma). D-Cycloserine (Alfa Aesar) and fosfomycin disodium (Sigma) were both stocked in sterile Dulbecco's PBS (Sigma) pH 7.2. Ciprofloxacin hydrochloride (Sigma) and bacitracin zinc (Sigma) were both stocked in sterile 0.1 M HCl.

### Extraction of plasmid and genomic DNA

Plasmid DNA was routinely extracted using the EZNA Plasmid DNA Mini Kit (Omega Bio-tek) and eluted in 10 mM Tris-HCl pH 8.5. Genomic DNA was isolated from *Burkholderia* with the PureLink Genomic DNA Mini Kit (Invitrogen) or by standard isopropanol precipitation and solubilized in 10 mM Tris-HCl pH 8.5.

## Construction, maintenance, and transfer of plasmids

The desired inserts were amplified by PCR using Q5 high-fidelity DNA polymerase (NEB) with the high-GC buffer as per the manufacturer's protocols. See Supplementary Table 9 for a list of primers used in this study. The Monarch PCR and DNA Cleanup Kit (NEB) and Monarch DNA Gel Extraction Kit (NEB) were routinely used to purify the DNA fragments. For restriction cloning, the appropriate restriction enzymes and T4 ligase (NEB) were used according to the manufacturer's recommendations to digest then ligate the vector and insert. To prevent self-ligation, vectors were treated with Antarctic phosphatase (NEB). For blunt-end ligation cloning, raw PCR product was mixed with T4 DNA ligase (NEB), DpnI (NEB), and T4 polynucleotide kinase (NEB) in a custom ligase buffer (132 mM Tris-HCl pH 7.5, 20 mM MgCl$_2$, 2 mM ATP, 15% PEG6000 in de-ionized water) and incubated for 30 mins at 37 °C. Expression plasmids were validated by long-read sequencing on Oxford Nanopore Technologies platforms by Plasmidsaurus (Oregon, USA).

A previously published python script was used to design sgRNAs[108]. sgRNAs were selected to bind as close as possible to the gene start, which has been shown to result in strong silencing[108]. To mitigate the effect of off-target gene silencing, sgRNAs with a 12 bp seed region matching any other site in the genome (adjacent to a PAM site) were removed. Additionally, sgRNAs were removed if they contained any of the following features suggested to reduce silencing efficiency: (1) PAM site of ANN or TNN, (2) the sgRNA ended in GG or GA or (3) very high (>80%) GC-content. Each gene was typically targeted by two different sgRNAs within the 5' most 75 bp, and the results of the mutant that displayed the stronger phenotype were reported. The silencing effect for each mutant was measured by qRT-PCR (see below) and is reported in Supplementary Table 2.

New sgRNA targeting regions were introduced into pSCB2-sgRNA by inverse PCR as previously described[46,47,109]. Briefly, the desired targeting sequence was added as a 5' extension to a primer with sequence: 5' – GTTTTAGAGCTAGAAATAGCAAGTTAAAATAAGGC – 3'. That primer was used with primer 1092 for PCR with Q5 DNA polymerase with high-GC buffer (NEB). The raw PCR product was used for blunt end ligation as above.

*E. coli* DH5α was used to maintain plasmids with broad-range origins of replication. *E. coli* SY327λpir+ was used to maintain plasmids containing the R6Kγ origin. Plasmids were introduced into these cells by standard transformation or electroporation.

Electro-competent *B. cenocepacia* K56-2 was made as per[110] with some modifications. Briefly, 5 mL of overnight culture of the appropriate strain was split into 1 mL aliquots and washed by centrifugation at 6 000 x g for 5 mins followed by resuspension in sterile, room temperature 300 mM sucrose in de-ionized water. This washing was repeated twice more and each of the aliquots were finally resuspended in 300 mM sucrose solution and used immediately.

More commonly, plasmids were introduced into Bcc species by triparental mating[29,47]. The *E. coli* donor strain, bearing the plasmid to be mobilized, the *E. coli* MM294/pRK2013 helper strain and the *Burkholderia* recipient were mated on an LB plate and incubated overnight (16–18 h) at 37 °C. Successful exconjugants were selected on LB plates containing the appropriate antibiotics with 50 μg mL$^{-1}$ gentamicin to select against the *E. coli* donor and helper. Colonies that appeared within 48 h at 37 °C were screened by colony PCR.

## Unmarked gene deletion in K56-2

Genes were deleted as per a double homologous recombination method reported previously[111,112], with some modifications. Briefly, 360 – 450 bp of the upstream and downstream regions flanking the gene(s) to be deleted were fused and ordered as a single gBlock from IDT with *Xma*I and *Xba*I restriction sites (see Supplementary Table 10 for all gBlock fragments). The fragments were cloned into pGPI-SceI and transformed into *E. coli* SY327. Plasmids were confirmed by colony PCR with primers 153 and 154, then introduced into K56-2::dCas9 by

triparental mating. The first recombination was confirmed by colony PCR of trimethoprim-resistant exconjugants. To promote the second recombination, pDAI-SceI-SacB[112] was introduced by triparental mating. Trimethoprim-sensitive colonies were screened by colony PCR for successful gene deletion. To cure pDAI-SceI-SacB, colonies were patched on LB (without NaCl) + 15% sucrose. Tetracycline-sensitive colonies were again screened by colony PCR for the deletion.

## Construction of the barcoded plasposon and transposon mutant library

We wished to model our approach to barcoding on a previous report from the Deutschbauer lab[21]. They constructed a 20 bp random barcode flanked by unique priming sites (GTCGACCTGCAGCGTACG and GAGACCTCGTGGACATC). These sequences are not found in K56-2 and were thus used for our approach. Primers 964 and 965 were used for inverse PCR of our parental plasposon pRBrhaBout. The product was circularized with In-Fusion HD Cloning (Takara Bio) as per the manufacturer's protocols. The resulting plasmid, pKS1, was sequence-confirmed to contain *Bam*HI and *Nco*I restriction sites between the inverted repeat and *dhfr*. A dsDNA fragment containing 20 random bases flanked by unique priming sites[21] and *Bam*HI and *Nco*I restriction sites was ordered from IDT with the sequence: 5'- GTTCAA*CCATG*-**GATGTCCACGAGGTCTCT**NNNNNNNNNNNNNNNNNNNN**CGTACGC****TGCAGGTCGAC***GGATCC*ACTTA - 3'. Italics show the *Nco*I and *Bam*HI restriction sites, while bold shows the unique priming sites; note that the end of the *Nco*I site (CCATGG) overlaps by one base (G) with one of the priming sites. As the fragment is used to generate the barcoded plasposon, pRBrha-Barcode, primers 972 and 973 were designed to amplify the fragment by PCR with Q5 polymerase and high-GC buffer (NEB). The barcode fragment and pKS1 were both digested with *Nco*I and *Bam*HI and ligated with T4 DNA ligase (NEB) at a 5:1 (insert:vector) ratio (100 ng total DNA per 20 μL reaction) overnight at 16 °C. The ligation mix was heat inactivated at 65 °C for 15 mins then 5 μL was electroporated into *E. coli* SY327 and plated on LB agar with 50 μg mL$^{-1}$ trimethoprim. Colonies were collected in subpools of 200 000 – 600 000 and frozen in multiple PCR tubes at -80 °C until needed. This process was repeated until ~6 million donor colonies were collected. A pilot BarSeq experiment (see below for the method) sequenced on a MiSeq with reagent kit v2 (Génome Québec, Montréal, Canada) using the first collected subpool confirmed it was highly diverse as expected.

Each subpool of *E. coli* was used only once as donor for triparental mating with K56-2. After only 4 h of mating at 37 °C (to reduce library redundancy), the mating pellicle was spread on Bioassay Qtrays (Molecular Devices) with the appropriate selective antibiotics and 0.2% rhamnose. After 2 days of growth, colonies were counted and harvested in subpools of 75 000 – 100 000 and frozen in PCR tubes at OD$_{600}$ 2.0 – 4.0 (~100 million CFU mL$^{-1}$, or ~100 CFU per mutant) at -80 °C. A pilot BarSeq experiment sequenced on a MiSeq with reagent kit v2 (Génome Québec, Montréal, Canada) was also performed on the first collected subpool of barcoded transposon mutants, which showed there were approximately half as many unique barcodes as total collected colonies.

## RB-TnSeq DNA library preparation and sequencing

The transposon-genome junctions were amplified by the TnSeq-circle method as previously reported[27,113], with some modifications to accommodate the barcoding scheme (RB-TnSeq-circle). Briefly, DNA was extracted from representatives of the entire transposon mutant library (not treated with antibiotics) by isopropanol precipitation. The DNA was sheared with a Covaris M220 ultrasonicator, followed by end repair with the NEBNext End Repair Module (NEB). Adaptors, consisting of annealed primers 683 and 684, were annealed with the Quick Ligation Kit (NEB). The DNA was then digested with *Pac*I (NEB) overnight. Probe 1426 in the presence of Ampligase (Lucigen) was used to

selectively circularize and protect fragments containing the barcoded transposon sequence, while all other fragments were digested with a mix of T7 *gene 6* exonuclease (ThermoFisher), Lambda exonuclease (NEB), and Exonuclease I (NEB). qPCR with iTaq Universal SYBR Green Supermix (Bio-Rad) was used to determine enrichment of transposon-genome junctions. The number of cycles corresponding to ~50% maximum product by qPCR was used to amplify transposon-genome junctions with iTaq for Illumina sequencing. All primers used for this purpose were constructed to enable 1-step PCR to add the Illumina flow cell adaptor, Nextera index, and sequencing primer binding site. After clean-up with SeraMag beads (Cytiva), the final PCR product was first analysed on a TapeStation4150 (Agilent Technologies) then sequenced on an Illumina MiSeq (Donnelly Centre, Toronto, Canada) to assess library quality with a MiSeq reagent kit v2 (2x250 bp reads) and 20% PhiX spike. To increase sequencing depth, an Illumina HiSeq 2500 (The Applied Centre for Genomics, Toronto, Canada) was used on rapid run mode with a 20% PhiX spike (1x150bp reads). In both cases, the sequencing reads covered the transposon-specific sequence, barcode region, and at least 40 bp of the genome adjacent to the insertion site. In total, ~180 million reads were generated. Sequencing platforms were operated with Illumina Experiment Manager.

### RB-TnSeq data analysis

To link barcodes to transposon insertion sites, the bioinformatic pipeline and scripts reported by Wetmore et al.[21] were used in a Conda 4.10.0 environment. These are available at https://bitbucket.org/berkeleylab/feba/src/master/. Briefly, the scripts identify reads containing the flanking priming sites and a transposon-specific sequence (terminal repeat) then map it to a genome with BLAT[114]. The priming sites flanking the barcode were the same as in the original report[21]. The transposon-specific sequence in our construct is CATGGG-CACTTGTGTATAAGAGTCAG. The default stringency and base error tolerances were used, resulting in 86.1% of the observed barcodes being considered "usable" (primarily associated with one insertion site and filtered to remove sequencing/PCR errors). The reads were aligned to the closed K56-2 genome (RefSeq accession GCF_014357995.1)[115].

### Transposon mutant pool antibiotic exposure

Wild-type K56-2 was used to determine the dose-response curves of the antibiotics in conditions mimicking the planned mutant pool exposure. Stationary phase cells were diluted to $OD_{600}$ 0.025 in LB medium and grown at 37 °C with 230 rpm shaking until early exponential phase ($OD_{600}$ 0.15, about 3 h). In 2 mL culture in glass tubes, a range of antibiotic concentrations was added (in technical duplicate) and the cells were grown for 8 h (~10–12 generations), again at 37 °C with 230 rpm shaking. Then, the $OD_{600}$ of each tube was measured to compute the dose-response curve for each antibiotic relative to cells grown without antibiotics. This assay was performed in three biological replicates. The $IC_{20-30}$ was confirmed by repeating this assay with a much narrower antibiotic concentration range around the $IC_{20-30}$.

To prepare the mutant inoculum, aliquots of the transposon mutant library were thawed and mixed in ratios according to the number of colonies contained in each aliquot to obtain approximately equal abundance of each of the ~340 000 unique mutants. This master pool was then inoculated into flasks with 50 mL of LB medium with 0.2% rhamnose at $OD_{600}$ 0.025 (~75 CFU mutant$^{-1}$) and grown at 37 °C with 230 rpm shaking until early exponential phase ($OD_{600}$ 0.15, about 5–6 h from frozen stocks). The culture was then split up into 2 mL volumes in small glass tubes, six for each antibiotic: three replicate tubes at two slightly different concentrations to ensure at least one achieved the $IC_{20-30}$. A 2 mL aliquot of cells not exposed to antibiotic was taken as the Time 0 sample. The cultures were then exposed to antibiotics (or 1% DMSO solvent control) for 8 h at 37 °C with 230 rpm shaking, after which the $OD_{600}$ of the cultures were taken and cells were harvested from the tubes at the $IC_{20-30}$.

### BarSeq DNA library preparation and sequencing

The method to amplify barcodes for sequencing followed a 1-step PCR as per[21] with some modifications. After antibiotic exposure, genomic DNA was extracted from cells with the PureLink Genomic DNA Mini Kit (Invitrogen). Yield for this and all subsequent steps was quantified with a Qubit (ThermoFisher) with either AccuGreen broad-range or high-sensitivity dsDNA detection kits (Biotium). To preserve mutant abundance ratios and reduce the propagation of PCR errors, semi-quantitative PCR was performed. This was done for six randomly selected antibiotic conditions. In each of a set of 10 μL reaction volumes, we added 40 ng of genomic DNA and 20 μM of each primer with Q5 high-fidelity DNA polymerase, the high-GC buffer, and standard Q5 reaction buffer (NEB). In an Eppendorf Mastercycler EP Gradient S, the cycling conditions were: 98 °C for 4 mins, followed by cycling between 98 °C for 30 s, 61 °C for 30 s, 72 °C for 30 s, then followed by a final extension at 72 °C for 5 mins. After each cycle between 13 and 21, we quickly removed one tube and added 3 μL of Gel Loading Dye, Purple 6x (NEB) (with 10 mM EDTA) to stop the reaction. Formation of the expected product at 196 bp was visualized by electrophoresis through a 2.5% agarose gel in standard TAE buffer. By visual inspection of band intensities, a cycle number was chosen that resulted in ~25% of maximum product formation. This cycle number (usually 17) was the same for all six randomly selected conditions and was thus used for all conditions. These cycling conditions were found in a small pilot experiment, sequenced on a MiSeq with reagent kit v2 and 20% PhiX spike, to result in 96.4% of reads containing known barcodes.

We found that amplification profiles in 10 μL volumes matched that in 50 μL, thus the exact same PCR setup was used when scaled up to four tubes of 50 μL per antibiotic condition to increase yields. For the template, 200 ng DNA was added to ensure >75 molecules of genome per mutant per tube. We observed a minor secondary product (<10%) at 315 bp on TapeStation 4150 traces (Agilent Technologies). Thus, for each condition, 200 μL of raw BarSeq PCR product was pooled and subjected to two rounds of dual size selection with Sera-Mag Select (Cytiva) magnetic beads to purify the desired product at 196 bp. The primers were designed with Nextera-type tagmentation sequences as for the RB-TnSeq-circle sequencing primers, except that the 8 bp standard Nextera indexes were replaced with 10 bp Unique Dual Indexes (primers 2163 – 2255, Table 3). Each product was amplified with a unique i5 and i7 index, enabling greater multiplexing flexibility and higher confidence in correcting up to 2 bp errors during indexing read sequencing. Up to 24 samples were indexed together for runs of a NextSeq 550 in high-output mode (Donnelly Centre, Toronto, Canada) with reagent kit v2.5 and 20% PhiX spike, generating 410–510 million 30 bp single-end reads each. A custom sequencing recipe was used for dark-cycling during the first 18 bases, covering the flanking primer region, with the read output starting at the beginning of the barcode and extending 10 bp into the other flanking priming region.

### BarSeq gene fitness calculations

BarSeq reads were associated with the correct barcode using the pipeline from[21]. Reads, in fastq format, were first trimmed to contain only the 20 bp barcode using the FASTX toolkit (http://hannonlab.cshl.edu/fastx_toolkit/). Reads with any base having a quality score below 20 were also filtered out. Artificial pre- and post-sequences, NNNGTCGACCTGCAGCGTACG and AGAGACC, respectively, were added to all barcodes so the script RunBarSeq.pl would recognize the barcodes as valid.

Following this, the barcodes were processed with scripts from[33] which are designed to compare mutant abundance between conditions (https://github.com/DuttonLab/RB-TnSeq-Microbial-interactions). A pseudocount of 0.1 was added to all barcode counts to prevent 0-value errors with log transformations. Barcodes were removed if they (1) had <3 reads in the Time 0 condition and (2) represented intergenic insertions, or insertions in the first or last 10% of a gene. Raw counts

were normalized against the counts of 10 non-essential "neutral" genes randomly selected from across the genome that also showed no fitness effect in any antibiotic condition. These were: K562_RS24650 (ABC-type amino acid transporter), K562_22855 (putative glycosyltransferase), K562_05000 (hydrolase family protein), K562_12100 (acyl-CoA dehydrogenase), K562_01045 (Raf kinase inhibitor-like protein), K562_06455 (putative PHA depolymerase protein), K562_13470 (GudD glucarate dehydratase), K562_16220 (DUF3025 domain-containing protein), K562_18550 (hypothetical protein), K562_28510 (hypothetical protein). Strain fitness values were then calculated as the $\log_2$(reads in Condition X/reads in Time 0). Gene fitness values were calculated as the arithmetic mean of strain fitness values for each gene. To account for artificial inflation of read count due to proximity to the replication forks, fitness values were smoothed based on genomic position using a moving window[21]. Across the three replicates, gene fitness was calculated as the inverse-variance weighted mean. Spearman, Pearson, and Lin correlation coefficients across replicates for each condition were between 0.5 and 0.8. Lastly, the gene fitness values in each condition were compared to the DMSO control by an independent two-sided Student's $t$-test. The data was also curated to remove any genes from downstream analysis that only had one reporting mutant. We used eggNOG-mapper v2 to provide functional annotations and gene names based on orthology[116,117].

As many bacterial genes are organized in operons, disruptions in upstream genes may have polar effects on the downstream gene expression. Consequently, observed effects would be a combination of disrupting two or more genes. We argue that polarity may have a minimal effect on our findings due to (1) high insertion density and (2) the construction of our transposon. The large number of mutants recovered in our screen virtually guarantees insertions in every non-essential gene, thus genomic saturation would rule out polar effects by reporting fitness scores for nearly all non-essential genes. Secondly, our transposon has outward-facing promoters in both directions: the rhamnose-inducible promoter and the promoter controlling *dhfr*. As the transposon mutant antibiotic exposures were performed with added rhamnose, and the transposon terminal repeat after *dhfr* is not predicted to form an intrinsic terminator, insertion in operons may permit downstream gene expression. The parent transposon pTn*Mod*-OTp[118] was suggested to not cause polar effects[119]. However, any expression, either upstream or downstream of the insertion, would be altered/non-native but this was not directly assessed.

## Microscopy

To observe the effect of deletion or overexpression of O-antigen-related genes, the appropriate mutants were inoculated in 2 mL glass tubes at $OD_{600}$ 0.025 with or without rhamnose (0.05%) and grown at 37 °C with 230 rpm shaking for 3 h. Cells were then diluted to $OD_{600}$ ~ 0.05, fixed in 3.7% formaldehyde + 1% methanol (Sigma) at room temperature for 20 min. The formaldehyde was quenched by addition of an equal volume of 0.5 M glycine, and the cells were washed by centrifugation and resuspended in Dulbecco's PBS (Sigma). Cells were spotted on 1.5% agarose pads with PBS to maintain turgor and imaged by DIC microscopy on an Axio Imager upright microscope operated with Zen 3.2 (Carl Zeiss Microscopy GmbH).

## LPS analysis

The general structure of the O-antigen LPS and lipid A core were analysed qualitatively by silver staining polyacrylamide gels as per[120], with some modifications. For complementation, cells were sub-cultured to $OD_{600}$ 0.025 and grown for 3 h, then induced with 0.05% rhamnose for an additional 4 h. For both types of mutants, an equivalent amount of 500 µL of $OD_{600}$ 1.5 was harvested, washed by centrifugation, and resuspended in 150 µL lysis buffer (2% SDS, 4% β-mercaptoethanol, 0.5 M Tris, pH 6.8). Cells were lysed by incubation at 95 °C for 10 min. DNase I (Worthington Biochemical) was added to

100 µg mL$^{-1}$ and incubated at 37 °C for 30 min. Proteinase K (Thermo-Fisher) was added to 1.25 mg mL$^{-1}$ and incubated at 60 °C for 2 h. An equal volume of 90% phenol (ThermoFisher) (supplemented with 0.1% β-mercaptoethanol and 0.2% 8-hydroxyquinoline) was added and the tubes were incubated at 70 °C for 15 min, with vortexing every 5 min. The extractions were chilled on ice for 10 min then centrifuged at 16,000 x g for 10 mins. The aqueous phase was washed with 10x volumes of ethyl ether (saturated with 10 mM Tris and 1 mM EDTA, pH 8.0) and centrifuged briefly. The top organic phase was removed by aspiration, then the aqueous phase was gently heated to 55 °C for 10 min to boil off the remaining ethyl ether. An equivalent volume of 2x loading dye (120 mM Tris-HCl pH 6.8, 4% SDS, 20% glycerol, 0.04% bromophenol blue, and 10% β-mercaptoethanol) was added and the samples were stored at −20 °C until needed.

The LPS banding patterns were resolved by Tricine-SDS-PAGE as per[120], with some modifications. The gels were cast using 30% acrylamide/bis 29:1 solution (Bio-Rad) at final concentrations of 14% for the resolving gel and 4% for the stacking gel. 10 uL of each sample, diluted with an equal volume of loading buffer, was run in each lane. Gels were run at 50 V for 1 hr, then at 130 V for 3 hrs and 15 mins, then stained with the Pierce Silver Stain Kit (ThermoFisher) and imaged immediately afterwards on a Bio-Rad ChemiDoc operated with Bio-Rad Image Lab.

## Rhamnose dose-response curves

After construction, the first objective with the CRISPRi and over-expression mutants was to characterize how varying gene knockdown or overexpression affects growth. Each of the mutants were diluted to $OD_{600}$ 0.01 in 96-well format and inoculated into fresh LB with a 2-fold rhamnose gradient (with 100 µg mL$^{-1}$ trimethoprim). Growth was monitored in a BioTek Synergy 2 plate reader, operated with Gen5 software, with constant linear shaking and temperature maintained at 37 °C.

## Antimicrobial susceptibility testing and interaction assays

Broth microdilution susceptibility tests were used to assess and interpret MIC values as per CLSI guidelines[121]. Overnight cultures grown in liquid LB were used to prepare inocula for standard micro-dilution tests and checkerboard interactions assays. Growth was visually assessed in the antibiotic gradients after 20 h of static incubation. When interpretive standards for *Burkholderia* did not exist (as for cefiderocol and aztreonam), standards for *Pseudomonas aeruginosa* were used.

A similar format to the standard MIC method above was used to assess changes in antibiotic susceptibility of the CRISPRi mutants. A final concentration of 0.5% rhamnose was added to the wells to induce dCas9 expression. For fitness-associated/essential genes (e.g. *uppS*), a concentration of rhamnose to achieve 80% WT growth was used. Tri-methoprim was not included in the assay plates to remove the possibility of chemical interactions. Growth was assessed after 20 hrs of static incubation by reading $OD_{600}$ values with a BioTek Synergy 2 plate reader to record subinhibitory dose effects.

Checkerboard assays were prepared in mini 4x4 well format using antibiotic concentrations of 1/8–1/2 the MIC for each compound. Testing was performed in cation-adjusted MHB as per CLSI guidelines. For testing with BAC, CAMHB buffered to pH 7.2 with 100 mM PBS was used as we found the acidic solvent for BAC antagonised β-lactam susceptibility (data not shown). Checkerboard data was processed with SynergyFinder2 (http://www.synergyfinder.org/#!/)[122,123] as this easily handles biological replicates and gives mathematical interpretations of interactions even when compounds do not fully inhibit. When both the Loewe Additivity and Bliss Independence scores were below -10 (as determined from the 95% confidence intervals), the interaction was regarded as strongly antagonistic; synergy was interpreted when both scores were above +10 (as determined from the 95% confidence

intervals). A weak interaction was interpreted when one score was above +10 or below -10 and the other was between -10 and +10.

### Inductively coupled plasma mass spectrometry (ICP-MS) analysis of trace metals in growth media

Samples of growth medium were prepared and autoclaved as per manufacturer's recommendations. The samples were clarified by passing through a 0.22 μm PVDF syringe filter (Fisher Scientific). Bound ions were liberated by room temperature digestion 1:1 with aqua regia. The solution was then diluted 20-fold with 18.2 MΩ cm$^{-1}$ water. ICP-MS analysis was carried out at the Ultra Clean Trace Elements Laboratory at the University of Manitoba with an Agilent 8900 ICP-QQQ-MS. The instrument plasma mode was set to general purpose with -140V omega bias and 6.6 V omega lens settings. $H_2$ gas flow was set to 4.5 mL min$^{-1}$; He gas flow was set to 4.3 mL min$^{-1}$; $O_2$ gas was set at 30%. Before the run, certified reference materials NIST-SRM 1640a and 1643e were used as a measure of quality control.

### NPN uptake assay

Outer membrane permeability was assessed with the NPN uptake assay as per[48,49], with some modifications. For CRISPRi mutants, overnight cultures were grown with or without 0.5% rhamnose (or 0.04% for the *uppS* mutants) then diluted to $OD_{600}$ 0.025 and grown for an additional 8 h (with or without the same concentration of rhamnose) until $OD_{600}$ ~ 1.0. Cells were washed by centrifugation at 5 000 x g for 5 min then resuspended in 5 mM HEPES pH 7.2 with 10 mM NaN$_3$ (Sigma) and depolarized at room temperature for 30 min. Then, cells were mixed with an equal volume of 5 mM HEPES buffer containing 80 μM NPN (Sigma). The fluorescence signal was measured in a BioTek Synergy 2 plate reader with filter sets: Ex 360/40 nm and Em 420/40 nm. Wild-type K56-2 treated with 32 μg mL$^{-1}$ CHX was used as a positive permeabilized control. The blank-corrected fluorescence values were normalized to the $OD_{600}$ of each well. To easily compare between replicates, the fluorescence ratios of the mutants to K56-2 controls (with non-targeting sgRNA as appropriate) were calculated.

### Nitrocefin hydrolysis assay of β-lactamase activity

This assay was performed as per[68,124] with some modifications. Overnight cultures of the selected CRISPRi mutants were prepared with and without 0.5% rhamnose. Cultures were diluted to $OD_{600}$ 0.1 in fresh LB with 100 μg mL$^{-1}$ trimethoprim, both with or without 0.5% rhamnose, then grown at 37 °C with shaking until $OD_{600}$ 2.0. Two mL of each culture were harvested, washed by centrifugation, and resuspended in lysis buffer (sodium phosphate pH 7.0, 1 mM EDTA, 40 μg mL$^{-1}$ lysozyme). Samples of CRISPRi mutants of the putative metallo-β-lactamase K562_RS32470 were lysed without EDTA. AVI was added to half of the samples at 8 μg mL$^{-1}$. Cells were rested at room temperature for 30 min then freeze-thawed rapidly at −80 °C. Cells were further lysed by probe sonication. The samples were clarified by centrifugation for 10 min at 13,000 x g at 4 °C and the supernatant was retained.

An equal volume of cell supernatant was mixed with a nitrocefin dilution in phosphate buffer pH 7.0 (final concentration of 100 μM nitrocefin (Sigma)). The absorbance at 485 nm was recorded every minute on a BioTek Synergy 2 plate reader. For each sample, the slope of the linear region of $A_{485}$ vs. Time was calculated. The specific activity of each sample was determined by normalizing the nitrocefin hydrolysis rate by the protein concentration of each sample, as determined by a BCA assay (Pierce).

### RNA extraction and qRT-PCR analysis of gene expression

Assays were performed as previously reported[47], with some modifications. Overnight cultures of the CRISPRi mutants, grown with 0.5% rhamnose (or 0.04% the mutant of *uppS*), were subcultured at $OD_{600}$ 0.01 for 8 h at 37 °C with rhamnose. Cells were harvested by centrifugation and the pellet was frozen at −80 °C for at least 3 h. Total RNA

was extracted with the PureLink RNA Mini kit (Invitrogen) and integrity was verified by running on a 2% agarose gel. DNase treatment and cDNA synthesis were performed with ezDNase and the SuperScript IV VILO reverse transcriptase (Invitrogen). PowerTrack SYBR Green Mastermix (Invitrogen) was used for qRT-PCR on a StepOnePlus PCR system operated with StepOnePlus software (Applied Biosystems). Primers (Supplementary Table 11) were designed to target the latter 50% of each gene. Primer efficiency was assessed on cDNA serial dilutions, and 90−110% was considered acceptable. Relative expression was measured with the comparative $C_T$ method[125] and normalized to expression of the *rpoD* RNA polymerase sigma factor as a housekeeping gene.

### Statistical analysis

For RB-TnSeq and BarSeq data, the statistical analysis and normalizations built into the scripts were used[21,33]. Only the independent Student's *t*-test was applied to the BarSeq data comparisons, not the false-discovery rate multiple-testing correction as it was found to be too stringent; however, samples were only compared by *t*-test if they passed Fisher's F-test of equal variances (α = 0.002). As this was expected to produce false positives, only genes with fitness effects >0.5 or <−0.5 were considered for further analysis. To support these findings, we performed extensive follow-up validations using CRISPRi mutants for many of the effects we observed in the BarSeq data. Pearson's correlation with two-tailed *p*-values was used to assess the relationship between gene fitness values in AVI/CAZ and the single conditions in the combination. The NPN outer membrane permeability assay was analysed by 1-way ANOVA with a Dunnett's multiple comparison test, with K56-2::dCas9 bearing the non-targeting sgRNA (or without for the deletion mutants) set as the reference. β-lactamase assay data was compared by unpaired two-tailed *t*-tests and adjusted using Bonferroni's multiple testing correction.

### Reporting summary

Further information on research design is available in the Nature Portfolio Reporting Summary linked to this article.

## Data availability

Raw sequencing data is available from the NCBI Sequencing Read Archive (SRA) under the BioProject ID PRJNA859150. All gene fitness scores are available in a spreadsheet in Supplementary Data 1. Publicly available databases and servers were used for gene annotations: Bio-Cyc (https://biocyc.org/), EggNOG-mapper v2 (http://eggnog-mapper.embl.de/), GO (http://geneontology.org/), and UniProt (https://www.uniprot.org/). The sequence of the closed K56-2 genome is under RefSeq accession GCF_014357995.1 Source data are provided with this paper.

## Code availability

Scripts and tools used to process sequencing data can be found at http://hannonlab.cshl.edu/fastx_toolkit/, https://bitbucket.org/berkeleylab/feba/src/master/, and https://github.com/DuttonLab/RB-TnSeq-Microbial-interactions. The Shiny app graphical user interface is freely available at https://cardonalab.shinyapps.io/bcc_interaction_viewer/, and the source code is available at https://github.com/cardonalab/Shiny-Bcc-ATB-Viewer.

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

## Acknowledgements

This work was financially supported by a Discovery Grant from the Natural Sciences and Engineering Research Council of Canada (NSERC), a Basic Research Grant from Cystic Fibrosis Canada, and a Research Project Grant from the Canadian Institutes of Health Research (CIHR) to S.T.C.; A.M.H. was supported by a Vanier Canada Graduate Scholarship from CIHR; A.N. was supported by an Undergraduate Summer Research Award from NSERC; B.L. and A.B. were supported by the Fondo para la Investigación Científica y Tecnológica (FONCyT) Proyecto de Investigación Trianual PICT-2019-III-A, Categoría Raíces. The authors thank Debbie Armstrong and Feiyue Wang (University of Manitoba) for ICP-MS analysis; Chi Dang and Ruwani L. Wimalasekara (Ayush Kumar group; University of Manitoba) for advice regarding qRT-PC; Dr. Krisztina Papp-Wallace (Veterans Affairs Northeast Ohio Healthcare System and Case Western Reserve University) for advice on the biochemistry of peptidoglycan recycling; the Donnelly Centre (Toronto, Canada), The Centre for Applied Genomics (Toronto, Canada), and Génome Québec (Montréal, Canada) for sequencing services.

## Author contributions

S.T.C. and A.M.H. conceived the idea and design of the research; A.M.H., B.L., and A.P. designed and constructed the randomly-barcoded transposon mutant library; A.M.H. performed the chemogenomics experiments, analysed the sequencing data, performed targeted validation studies, and wrote and edited the manuscript; DTM aided with sequencing data analysis; A.S.M.Z.R., A.M.H., and A.M. performed and analyzed qRT-PCR experiments; A.N. constructed CRISPRi mutants related to cefiderocol activity; A.M. performed the lipopolysaccharide analyses; Z.B. assessed β-lactam susceptibility of Bcc clinical isolates; STC edited the manuscript, and together with A.B., supervised the work and provided financial support. A.M. and A.S.M.Z.R. contributed equally as second authors.

## Competing interests

The authors declare no competing interests
