## [Peer Review File · Nature Communications]

Profiling cell envelope-antibiotic interactions reveals vulnerabilities to β -lactams in a multidrug-resistant bacteriumReviewer #1 (Remarks to the Author):

This manuscript by Hogan et. al., describes a comprehensive mutant analysis of genes that confer fitness changes in response antibiotics that target the cell envelope of *Burkholderia cepacia* complex. The main takeaways were that saturating transposon mutagenesis combined with CRISPRi validation identified genes in the Mla pathway where reduced fitness scores were seen with almost all the antibiotics tested. Mutations in undecaprenol phosphate synthesis/recycling sensitized cells to beta-lactams. As seen in other bacteria, the loss of iron uptake pathways increased cefidericol resistance and this study revealed new iron receptors involved in cefiderocol uptake. The manuscript reports important new information and the data is presented in a comprehensive manner. I have the following suggestions and comments that includes one major concern.

1. For all the CRISPRi studies, the degree of reduced expression for each target gene was not reported This data should be determined and reported because without this data, the conclusions drawn about the roles of various genes involved in susceptibilities may be incorrect.
2. In Fig. 2C mutations in the *miaFED/vacJ* operon had a stronger fitness defect than those in *miaCB*. In contrast, a CRISPRi knockdown of *miaCB* has a stronger effect on permeability than that for *miaFED/vacJ* (Fig. 2E). Do the author's have an explanation for why the *miaFED/vacJ* knockdown was not as strong as *miaCB*? Also, please indicate what NTC designates in Fig. 2E and elsewhere.
3. As a general comment on mutations that result in the accumulation of UndP intermediates, these can also activate cell envelope stress regulatory pathways such as the Rcs phosphorelay and these may be contributing to some of the observed phenotypes.
4. The idea that an *ampC* knockdown increases PenB expression is interesting and the study would be strengthened by directly testing this possibility by qRT-PCR and determining if the *ampC* knockdown exerts it effects through PenR (lines 547-549).
5. Line 413: I think Figure 6B should be 6A.
6. Line 415: Please indicate the function of PiuA

Reviewer #2 (Remarks to the Author):

The *Burkholderia cepacia* complex (Bcc) is a group of Gram-negative bacteria composed of more than 20 species that cause pneumonia in immunocompromised individuals. Bcc infections are notoriously difficult to treat and many strains exhibit resistance to several front-line antibiotics simply because they cannot pass through the outer membrane. Thus, there is a pressing need to identify factors that mediate cell-envelope associated resistance in the Bcc. Prompted by these concerns, Hogan et al. generated a randomly-barcoded transposon mutant library in the clinical isolate *Burkholderia cenocepacia* K56-2 and screened for mutants exhibiting increased sensitivity to cell envelope-targeting antibiotics. The screen was highly successful, corroborating several previous observations, as well as identifying many new functional associations. The authors used their mutant library to show (1) how a functioning Mla pathway excludes large antibiotics from passing the outer membrane, (2) how disruptions in Und-P metabolism increase susceptibility to beta-lactams, and (3) evidence that PenB is the predominant beta-lactamase in K56-2. The authors also show how low iron potentiates the activity of Cefiderocol and that TonB-dependent receptors are likely required for Cefiderocol to cross the outer membrane.

Overall, the manuscript is well-written, the experiments well-controlled, and the conclusions (mostly) appropriate. The inclusion of a searchable database for the chemical genomics data is a very helpful addition. The construction of barcoded library will also be a helpful addition to those

working on Burkholderia and will likely be used to identify additional gene-drug interactions. I have minor comments.

Line 366: "...we suggest that knockdown of blaAmpC may induce overexpression of blaPenB." Can the authors simultaneously knock down blaAmpC and blaPenB and check for beta-lactam susceptibility?

Lines 302-303: "Upon induction of dCas9 with rhamnose, knockdown of ispDF and uppS resulted in a double mutant effect with a further decrease in growth..." Knocking down ispDF or uppS reduces growth (Fig. 5A) but the effects of Und-P sequestration do not appear to change the magnitude of difference between untreated (NTC) and treated strains (pgRNA).

Lines 513-514: "Additionally, O-antigen synthesis in Burkholderia may require less UndP than in E. coli." The suggestion that less UndP is required to assemble O-antigen polymers via an ABC transporter-dependent pathway versus a Wzy-dependent pathway is interesting. However, E. coli O8 and O9 are also synthesized via an ABC transporter-dependent pathway and mutants produce shape defects (PMID: 15980069). It may be that the unique characteristics of the Burkholderia cell envelope suppress the effects of Und-P sequestration?

Line 526: "DbcA, a DedA-family UndP flippase and homologue of UptA in E. coli..." To help avoid confusion, UptA is the Und-P flippase in Bacillus subtilis. DedA is the UptA homolog in E. coli.

Lines 540-541: Did the authors compare their chemical genomics data to any other large-scale studies (e.g., PMID: 21185072)?

Other

Lines 314-315: "lack of UndP" to "Lack of UndP"

Line 426: "Contrary what was" to "Contrary to what was"

Line 686: "The ligation mix was heated inactivated" to "The ligation mix was heat inactivated"

We thank the reviewers for their time and effort in reading our manuscript. We have carefully considered the comments and prepared thoughtful responses, edited the manuscript, and performed additional experiments, where appropriate.

Please note: Line numbers refer to the marked-up version of the manuscript, and changes are **highlighted in yellow**.

Reviewer 1

1. For all the CRISPRi studies, the degree of reduced expression for each target gene was not reported. This data should be determined and reported because without this data, the conclusions drawn about the roles of various genes involved in susceptibilities may be incorrect.

We agree that this is an important validation. We have confirmed reduced target gene expression by qRT-PCR in most of CRISPRi mutants used in this study (except for penB and penR, see below). Each of the mutants displayed between 1.1- and 79.4- fold reduction in gene expression upon rhamnose induction of CRISPRi. We have reported these findings in Supplementary Table 2; Supplementary Table 12 was created to show qRT-PCR primers and their efficiency.

Additionally, we have added a new section to the Methods and Materials describing these assays and, in the Methods section describing sgRNA plasmid creation, reported that qRT-PCR was used for mutant validation. Please see lines 653-5 and 976-89. We have also added new acknowledgements on lines 1368-70 and re-arranged the author list.

Despite the strong gene silencing effect of CRISPRi, the qRT-PCR experiment only showed minor effects of silencing penB and penR in CAMHB (1.1-1.6 fold repression), and their expression was near the lower limit of quantification. These genes are known to be upregulated by β -lactam exposure, so we assume that our inability to detect changes in expression was due to the absence of β -lactams in the growing conditions. We have included a statement of limitation in our discussion, please see lines 555-7. In the current manuscript, we are confident in the involvement of penB and penR in antibiotic susceptibility as this is supported by the findings from an orthogonal assay (nitrocefinase activity).

Finally, we opted not to perform qRT-PCR on the mutants related to TBDRs. In other species, expression of genes related to iron acquisition depends strongly on the growth conditions. Little is known in Burkholderia, let alone if qRT-PCR can accurately quantify their foreseeably low expression in iron replete conditions (i.e. rich medium). As the panel of TBDR CRISPRi mutants is expected to be a useful resource, we wish to save a fuller characterization of these mutants for future publication.

2. In Fig. 2C mutations in the mlaFED/vacJ operon had a stronger fitness defect than those in mlaCB. In contrast, a CRISPRi knockdown of mlaCB has a stronger effect on permeability than

that for mlaFED/vacJ (Fig. 2E). Do the author's have an explanation for why the mlaFED/vacJ knockdown was not as strong as mlaCB? Also, please indicate what NTC designates in Fig. 2E and elsewhere

The reviewer is correct that this appears to be a discrepancy. The difference is due to the types of mutants used. The heatmap in Figure 2C reports on transposon mutants with inactivated genes, while Figure 2E reports on CRISPRi mutants. In our qRT-PCR experiment of the CRISPRi mutants, we found that mlaCB was repressed by 17.8-fold, while mlaFEDvacJ was repressed by only 3.6-fold in their respective mutants. Thus, we expect the stronger permeability effect with mlaCB in Figure 2E is due to a stronger knockdown. We have indicated this in the text, please see lines 218-20..

Additionally, thank you for pointing out the issue with the captions. We have ensured the NTC abbreviation is clear in this and all other captions.

3. As a general comment on mutations that result in the accumulation of UndP intermediates, these can also activate cell envelope stress regulatory pathways such as the Rcs phosphorelay and these may be contributing to some of the observed phenotypes.

The reviewer raises a good point. Unfortunately, while these pathways have been characterised in model organisms, envelope stress responses are not well studied in Burkholderia species. Only the RpoE/extracytoplasmic sigma factor response has been studied in Burkholderia (Flannagan and Valvano, 2008; doi: 10.1099/mic.0.2007/013714-0). Searching by BLAST, the K56-2 genome lacks clear homologues of the Rcs and Cpx systems.

We reasoned that if accumulation of UndP intermediates activated elements of the RpoE pathway in K56-2, then disruptions in this pathway would result in altered antibiotic susceptibility (especially to the β -lactams) in the BarSeq experiment. Reviewing our data, we only observed significant interactions between the RpoE pathway and the membrane disrupting antibiotics polymyxin B and chlorhexidine, suggesting there is no direct link between UndP and the RpoE pathway in K56-2. However, it is still possible that other unknown envelope stress pathways in K56-2 may sense and respond to levels of UndP intermediates. We have included some of these points in the discussion, please see lines 503-13.

4. The idea that an ampC knockdown increases PenB expression is interesting and the study would be strengthened by directly testing this possibility by qRT-PCR and determining if the ampC knockdown exerts its effects through PenR (lines 547-549).

Indeed, we also find this link intriguing. As recommended, we used qRT-PCR in the bla_{AmpC} CRISPRi mutant to assess expression of penR, bla_{AmpC}, bla_{PenB}, and K562_RS32470. The results are reported in Supplementary Figure 13B and on lines 368-71. As we expected, there was a large increase in bla_{PenB} expression (9.1-fold) but no change in expression of penR, or

K562_RS32470. This supports our idea that AmpC may play an additional regulatory role in the response to β -lactams.

5. Line 413: I think Figure 6B should be 6A.

Thank you for the correction, we have made the change on line 420.

6. Line 415: Please indicate the function of PiuA

We have made the suggested addition on line 422.

Reviewer 2

Line 366: "...we suggest that knockdown of blaAmpC may induce overexpression of blaPenB." Can the authors simultaneously knock down blaAmpC and blaPenB and check for beta-lactam susceptibility?

This is an important point also raised by reviewer #1 in comment #4. Instead of creating a multiplex CRISPRi mutant, we used qRT-PCR to directly quantify expression of other β -lactamases and associated genes in the bla_{AmpC} CRISPRi mutant. Only expression of bla_{PenB} was notably changed (upregulated ~9.1-fold vs. the control CRISPRi mutant), and is reported in Supplementary Figure 13B and in lines 368-71. We believe these findings strength our assertion that AmpC also has an important regulatory role.

Lines 302-303: "Upon induction of dCas9 with rhamnose, knockdown of ispDF and uppS resulted in a double mutant effect with a further decrease in growth..." Knocking down ispDF or uppS reduces growth (Fig. 5A) but the effects of Und-P sequestration do not appear to change the magnitude of difference between untreated (NTC) and treated strains (pgRNA).

The reviewer is correct in pointing out that the absolute difference in OD did not change notably when knocking down uppS and ispDF in Figure 5A.

As an alternative, when presented as a relative proportion to the starting OD, the differences in the amount of growth at increasing concentrations of rhamnose for the mutants are now visible compared to the control strains. In chemical genetics studies, reporting growth as relative/normalized to starting/control conditions is common practice (e.g. PMID: 32156814, 36098580, 31924499 and 31451773). At other points in our manuscript (Figure 3D and 5D) we also represent growth as relative to a control condition.

We have thus remade Figure 5A and 5B by normalizing the growth at each concentration of rhamnose back to the growth at 0% rhamnose. We have also reordered the panels to be grouped by the CRISPRi knockdown or overexpression plasmid (rather than by genetic background).

Doing this emphasizes that the Δ waaL and Δ hldD mutants are indeed more susceptible to knockdown of ispDF and uppS, and to overexpression of wbiI and wzm-wzt. Additionally, showing the data in relative terms for Figure 5A and 5B adds continuity with how this type data is shown elsewhere in the manuscript. We have also made the appropriate changes to the figure caption, please see lines 1463-5..

Lines 513-514: “Additionally, O-antigen synthesis in Burkholderia may require less UndP than in E. coli.” The suggestion that less UndP is required to assemble O-antigen polymers via an ABC transporter-dependent pathway versus a Wzy-dependent pathway is interesting. However, E. coli O8 and O9 are also synthesized via an ABC transporter-dependent pathway and mutants produce shape defects (PMID: 15980069). It may be that the unique characteristics of the Burkholderia cell envelope suppress the effects of Und-P sequestration?

Thank you for bringing this study to our attention. We have added a reference to it on line 524 and 526 (reference #92).

We agree that multiple factors may result in the robustness of the Burkholderia cell envelope to UndP stress. We have removed the discussion about differences in the ABC vs. Wzy pathways as the effects were not clear cut given the findings in the paper the reviewer highlighted (PMID: 15980069). Additionally, we have added more nuance to the Burkholderia specific aspects of envelope physiology. We speculate that the Burkholderia-specific essentiality of the Ara4N modification on LPS may mean UndP is more abundant compared to other models (as each molecule of Ara4N requires one UndP carrier). We have addressed this by adding a new paragraph and details to a section in the discussion, see lines 525-34.

Line 526: “DbcA, a DedA-family UndP flippase and homologue of UptA in E. coli...” To help avoid confusion, UptA is the Und-P flippase in Bacillus subtilis. DedA is the UptA homolog in E. coli.

Our apologies for the mixup. We have double checked our homology data and fixed the statement accordingly. Please see lines 256 and 541.

Lines 540-541: Did the authors compare their chemical genomics data to any other large-scale studies (e.g., PMID: 21185072)?

We are very interested in the similarities and differences in antibiotic responses in different organisms. Prior screens in model organisms are indeed a valuable resource. A thorough comparison of our dataset to the screens already published is a large undertaking, and we are working with other scientists that have performed Tn-seq approaches to build an online comparative tool.

In the current manuscript, we highlighted key interactions from published literature when they arose to place our work into context. For example, we pointed out that others have seen β -lactam-UndP interactions before (lines 499-501) and that Tn mutagenesis can link antibiotics with potentiators (lines 558).

Other

Lines 314-315: “lack of UndP” to “Lack of UndP”

Line 426: “Contrary what was” to “Contrary to what was”

Line 686: “The ligation mix was heated inactivated” to “The ligation mix was heat inactivated”

We have addressed all of these typos, please see lines 318, 432, and 707.

Reviewer #1 (Remarks to the Author):

My concerns have been addressed and I have no additional comments